# Learning Restricted Boltzmann Machines
# with Sparse Latent Variables

**Guy Bresler**
MIT
guy@mit.edu

**Rares-Darius Buhai**[*]
MIT
rbuhai@mit.edu

## Abstract

Restricted Boltzmann Machines (RBMs) are a common family of undirected graphical models with latent variables. An RBM is described by a bipartite graph, with all observed variables in one layer and all latent variables in the other. We consider the task of learning an RBM given samples generated according to it. The best algorithms for this task currently have time complexity $\tilde{O}(n^2)$ for ferromagnetic RBMs (i.e., with attractive potentials) but $\tilde{O}(n^d)$ for general RBMs, where $n$ is the number of observed variables and $d$ is the maximum degree of a latent variable. Let the *MRF neighborhood* of an observed variable be its neighborhood in the Markov Random Field of the marginal distribution of the observed variables. In this paper, we give an algorithm for learning general RBMs with time complexity $\tilde{O}(n^{2^s+1})$, where $s$ is the maximum number of latent variables connected to the MRF neighborhood of an observed variable. This is an improvement when $s < \log_2(d-1)$, which corresponds to RBMs with sparse latent variables. Furthermore, we give a version of this learning algorithm that recovers a model with small prediction error and whose sample complexity is independent of the minimum potential in the Markov Random Field of the observed variables. This is of interest because the sample complexity of current algorithms scales with the inverse of the minimum potential, which cannot be controlled in terms of natural properties of the RBM.

## 1 Introduction

### 1.1 Background

Undirected graphical models, also known as *Markov Random Fields* (MRFs), are probabilistic models in which a set of random variables is described with the help of an undirected graph, such that the graph structure corresponds to the dependence relations between the variables. Under mild conditions, the distribution of the random variables is determined by potentials associated with each clique of the graph [11].

The joint distribution of any set of random variables can be represented as an MRF on a complete graph. However, MRFs become useful when the graph has nontrivial structure, such as bounded degree or bounded clique size. In such cases, learning and inference can often be carried out with greater efficiency. Since many phenomena of practical interest can be modelled as MRFs (e.g., magnetism [5], images [18], gene interactions and protein interactions [25, 8]), it is of great interest to understand the complexity, both statistical and computational, of algorithmic tasks in these models.

The expressive power of graphical models is significantly strengthened by the presence of latent variables, i.e., variables that are not observed in samples generated according to the model. However, algorithmic tasks are typically more difficult in models with latent variables. Results on learning

---

[*]Current affiliation: ETH Zurich, rares.buhai@inf.ethz.ch.

models with latent variables include [19] for hidden Markov models, [7] for tree graphical models, [6] for Gaussian graphical models, and [1] for locally tree-like graphical models with correlation decay.

In this paper we focus on the task of learning *Restricted Boltzmann Machines* (RBMs) [23, 9, 12], which are a family of undirected graphical models with latent variables. The graph of an RBM is bipartite, with all observed variables in one layer and all latent variables in the other. This encodes the fact that the variables in one layer are jointly independent conditioned on the variables in the other layer. In practice, RBMs are used to model a set of observed features as being influenced by some unobserved and independent factors; this corresponds to the observed variables and the latent variables, respectively. RBMs are useful in common factor analysis tasks such as collaborative filtering [21] and topic modelling [13], as well as in applications in domains as varied as speech recognition [14], healthcare [27], and quantum mechanics [20].

In formalizing the learning problem, a challenge is that there are infinitely many RBMs that induce the same marginal distribution of the observed variables. To sidestep this non-identifiability issue, the literature on learning RBMs focuses on learning the marginal distribution itself. This marginal distribution is, clearly, an MRF. Call the *order* of an MRF the size of the largest clique that has a potential. Then, more specifically, it is known that the marginal distribution of the observed variables is an MRF of order at most $d$, where $d$ is the maximum degree of a latent variable in the RBM. Hence, one way to learn an RBM is to simply apply algorithms for learning MRFs. The best current algorithms for learning MRFs have time complexity $\tilde{O}(n^r)$, where $r$ is the order of the MRF [10, 16, 24]. Applying these algorithms to learning RBMs therefore results in time complexity $\tilde{O}(n^d)$. We note that these time complexities hide the factors that do not depend on $n$.

This paper is motivated by the following basic question:

*In what settings is it possible to learn RBMs with time complexity substantially better than $\tilde{O}(n^d)$?*

Reducing the runtime of learning arbitrary MRFs of order $r$ to below $n^{\Omega(r)}$ is unlikely, because learning such MRFs subsumes learning noisy parity over $r$ bits [2], and it is widely believed that learning $r$-parities with noise (LPN) requires time $n^{\Omega(r)}$ [15]. For ferromagnetic RBMs, i.e., RBMs with non-negative interactions, [4] gave an algorithm with time complexity $\tilde{O}(n^2)$. In the converse direction, [4] gave a general reduction from learning MRFs of order $r$ to learning (non-ferromagnetic) RBMs with maximum degree of a latent variable $r$.

In other words, the problem of learning RBMs is just as challenging as for MRFs, and therefore learning general RBMs cannot be done in time less than $n^{\Omega(d)}$ without violating conjectures about LPN.

The reduction in [4] from learning order $r$ MRFs to learning RBMs uses an *exponential* in $r$ number of latent variables to represent each neighborhood of the MRF. Thus, there is hope that RBMs with *sparse* latent variables are in fact easier to learn than general MRFs. The results of this paper demonstrate that this is indeed the case.

## 1.2 Contributions

Let the *MRF neighborhood* of an observed variable be its neighborhood in the MRF of the marginal distribution of the observed variables. Let $s$ be the maximum number of latent variables connected to the MRF neighborhood of an observed variable. We give an algorithm with time complexity $\tilde{O}(n^{2^s+1})$ that recovers with high probability the MRF neighborhoods of all observed variables. This represents an improvement over current algorithms when $s < \log_2(d-1)$.

The reduction in time complexity is made possible by the following key structural result: if the mutual information $I(X_u; X_I | X_S)$ is large for some observed variable $X_u$ and some subsets of observed variables $X_I$ and $X_S$, then there exists a subset $I'$ of $I$ with $|I'| \leq 2^s$ such that $I(X_u; X_{I'} | X_S)$ is also large. This result holds because of the special structure of the RBM, in which, with few latent variables connected to the neighborhood of any observed variable, not too many of the low-order potentials of the induced MRF can be cancelled.

Our algorithm is an extension of the algorithm of [10] for learning MRFs. To find the neighborhood of a variable $X_u$, their algorithm iteratively searches over all subsets of variables $X_I$ with $|I| \leq d-1$ for one with large mutual information $I(X_u; X_I | X_S)$, which is then added to the current set of neighbors

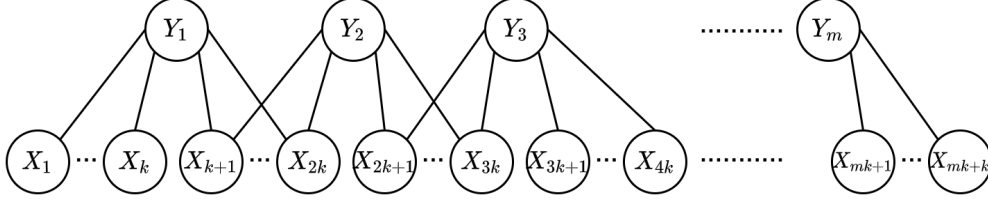

Figure 1: Class of RBMs with $mk + k$ observed variables, $m$ latent variables, $d = 2k$, and $s = 4$. The $X$ variables represent observed variables, the $Y$ variables represent latent variables, and the edges represent non-zero interactions between variables. The "$\cdots$" hides variables that have consecutive indices. The variables hidden by "$\cdots$" have the same connections as the variables at the extremes of their respective dots.

$X_S$. Our structural result implies that it is sufficient to search over subsets $X_I$ with $|I| \leq 2^s$, which reduces the time complexity from $\tilde{O}(n^d)$ to $\tilde{O}(n^{2^s+1})$.

For our algorithm to be advantageous, it is necessary that $s < \log_2(d-1)$. Note that $s$ is implicitly also an upper bound on the maximum degree of an observed variable in the RBM. Figure 1 shows an example of a class of RBMs for which our assumptions are satisfied. In this example, $s$ can be made arbitrarily smaller than $d$, $n$, and the number of latent variables.

The sample complexity of our algorithm is the same as that of [10], with some additional factors due to working with subsets of size at most $2^s$. We extended [10] instead of one of [16, 24], which have better sample complexities, because our main goal was to improve the time complexity, and we found [10] the most amenable to extensions in this direction. The sample complexity necessarily depends on the width (defined in Section 2) and the minimum absolute-value non-zero potential of the MRF of the observed variables [22]. In the Appendix F, we show that our sample complexity actually depends on a slightly weaker notion of MRF width than that used in current papers. This modified MRF width has a more natural correspondence with properties of the RBM.

The algorithm we described only recovers the structure of the MRF of the observed variables, and not its potentials. However, recovering the potentials is easy after the structure is known: e.g., see Section 6.2 in [4].

The second contribution of this paper is an algorithm for learning RBMs with time complexity $\tilde{O}(n^{2^s+1})$ whose sample complexity does not depend on the minimum potential of the MRF of the observed variables. The algorithm is not guaranteed to recover the correct MRF neighborhoods, but is guaranteed to recover a model with small prediction error (a distinction analogous to that between support recovery and prediction error in regression). This result is of interest because all current algorithms depend on the minimum potential, which can be degenerate even when the RBM itself has non-degenerate interactions. Learning graphical models in order to make predictions was considered before in [3] for trees.

In more detail, we first give a structure learning algorithm that recovers the MRF neighborhoods corresponding to large potentials. Second, we give a regression algorithm that estimates the potentials corresponding to these MRF neighborhoods. Lastly, we quantify the error of the resulting model for predicting the value of an observed variable given the other observed variables. Overall, we achieve prediction error $\epsilon$ with a sample complexity that scales exponentially with $\epsilon^{-1}$, and that otherwise has dependencies comparable to our main algorithm.

### 1.3 Overview of structural result

We present now the intuition and techniques behind our structural result. Theorem 1 states an informal version of this result.

**Theorem 1** (Informal version of Theorem 4)**.** *Fix observed variable $u$ and subsets of observed variables $I$ and $S$, such that all three are disjoint. Suppose that $I$ is a subset of the MRF neighborhood of $u$ and that $|I| \leq d-1$. Then there exists a subset $I' \subseteq I$ with $|I'| \leq 2^s$ such that*

$$\nu_{u,I'|S} \geq C_{s,d} \cdot \nu_{u,I|S}$$

*where $C_{s,d} > 0$ depends on $s$ and $d$, and where $\nu_{u,I',S}$ and $\nu_{u,I|S}$ are proxies of $I(X_u, X_{I'}|X_S)$ and $I(X_u, X_I|X_S)$, respectively.*

The formal definition of $\nu$ is in Section 2. For the purposes of this section, one can think of it as interchangeable with the mutual information. Furthermore, this section only discusses how to obtain a point-wise version of the bound, $\nu_{u,I'|S}(x_u, x_{I'}|x_S) \geq C'_{s,d} \cdot \nu_{u,I|S}(x_u, x_I|x_S)$, evaluated at specific $x_u$, $x_I$, and $x_S$. It is not too difficult to extend this result to $\nu_{u,I'|S} \geq C_{s,d} \cdot \nu_{u,I|S}$.

In general, estimating the MRF neighborhood of an observed variable is hard because the low-order information between the observed variables can vanish. In that case, to obtain any information about the distribution, it is necessary to work with high-order interactions of the observed variables. Typically, this translates into large running times.

Theorem 1 shows that if there is some high-order $\nu_{u,I|S}$ that is non-vanishing, then there is also some $\nu_{u,I'|S}$ with $|I'| \leq 2^s$ that is non-vanishing. That is, the order up to which all the information can vanish is less than $2^s$. Or, in other words, RBMs in which all information up to a large order vanishes are complex and require *many* latent variables.

To prove this result, we need to relate the mutual information in the MRF neighborhood of an observed variable to the number of latent variables connected to it. This is challenging because the latent variables have a non-linear effect on the distribution of the observed variables. This non-linearity makes it difficult to characterize what is "lost" when the number of latent variables is small.

The first main step of our proof is Lemma 7, which expresses $\nu_{u,I|S}(x_u, x_I|x_S)$ as a sum over $2^s$ terms, representing the configurations of the latent variables connected to $I$. Each term of the sum is a product over the observed variables in $I$. This expression is convenient because it makes explicit the contribution of the latent variables to $\nu_{u,I|S}(x_u, x_I|x_S)$. The proof of the lemma is an "interchange of sums", going from sums over configurations of observed variables to sums over configurations of latent variables.

The second main step is Lemma 8, which shows that for a sum over $m$ terms of products over $n$ terms, it is possible to reduce the number of terms in the products to $m$, while decreasing the original expression by at most a factor of $C'_{m,n}$, for some $C'_{m,n} > 0$ depending on $n$ and $m$. Combined with Lemma 7, this result implies the existence of a subset $I'$ with $|I'| \leq 2^s$ such that $\nu_{u,I'|S}(x_u, x_{I'}|x_S) \geq C'_{s,d} \cdot \nu_{u,I|S}(x_u, x_I|x_S)$.

## 2 Preliminaries and notation

We start with some general notation: $[n]$ is the set $\{1, ..., n\}$; $\mathbb{1}\{A\}$ is 1 if the statement $A$ is true and 0 otherwise; $\binom{n}{k}$ is the binomial coefficient $\frac{n!}{k!(n-k)!}$; $\sigma(x)$ is the sigmoid function $\sigma(x) = \frac{1}{1+e^{-x}}$.

**Definition 2.** *A Markov Random Field[2] of order $r$ is a distribution over random variables $X \in \{-1, 1\}^n$ with probability mass function*

$$\mathbb{P}(X = x) \propto \exp(f(x))$$

*where $f$ is a polynomial of order $r$ in the entries of $x$.*

Because $x \in \{-1, 1\}^n$, it follows that $f$ is a multilinear polynomial, so it can be represented as

$$f(x) = \sum_{S \subseteq [n]} \hat{f}(S)\chi_S(x)$$

where $\chi_S(x) = \prod_{i \in S} x_i$. The term $\hat{f}(S)$ is called the Fourier coefficient corresponding to $S$, and it represents the potential associated with the clique $\{X_i\}_{i \in S}$ in the MRF. There is an edge between $X_i$ and $X_j$ in the MRF if and only if there exists some $S \subseteq [n]$ such that $i, j \in S$ and $\hat{f}(S) \neq 0$. Some other relevant notation for MRFs is: let $D$ be the maximum degree of a variable; let $\alpha$ be the minimum absolute-value non-zero Fourier coefficient; let $\gamma$ be the width:

$$\gamma := \max_{u \in [n]} \sum_{\substack{S \subseteq [n] \\ u \in S}} |\hat{f}(S)|.$$

**Definition 3.** *A* Restricted Boltzmann Machine *is a distribution over observed random variables* $X \in \{-1, 1\}^n$ *and latent random variables* $Y \in \{-1, 1\}^m$ *with probability mass function*

$$\mathbb{P}(X = x, Y = y) \propto \exp\left(x^T J y + h^T x + g^T y\right)$$

*where* $J \in \mathbb{R}^{n \times m}$ *is an interaction (or weight) matrix,* $h \in \mathbb{R}^n$ *is an external field (or bias) on the observed variables, and* $g \in \mathbb{R}^m$ *is an external field (or bias) on the latent variables.*

There exists an edge between $X_i$ and $Y_j$ in the RBM if and only if $J_{i,j} \neq 0$. The resulting graph is bipartite, and all the variables in one layer are conditionally jointly independent given the variables in the other layer. Some other relevant notation for RBMs is: let $d$ be the maximum degree of a latent variable; let $\alpha^*$ be the minimum absolute-value non-zero interaction; let $\beta^*$ be the width:

$$\beta^* := \max\left(\max_{i \in [n]} \sum_{j=1}^m |J_{i,j}| + |h_i|, \max_{j \in [m]} \sum_{i=1}^n |J_{i,j}| + |g_j|\right).$$

In the notation above, we say that an RBM is $(\alpha^*, \beta^*)$-consistent. Typically, to ensure that the RBM is non-degenerate, it is required for $\alpha^*$ not to be too small and for $\beta^*$ not to be too large; otherwise, interactions can become undetectable or deterministic, respectively, both of which lead to non-identifiability [22].

In an RBM, it is known that there is a lower bound of $\sigma(-2\beta^*)$ and an upper bound of $\sigma(2\beta^*)$ on any probability of the form

$$\mathbb{P}(X_u = x_u | E) \quad \text{or} \quad \mathbb{P}(Y_u = y_u | E)$$

where $E$ is any event that involves the other variables in the RBM. It is also known that the marginal distribution of the observed variables is given by (e.g., see Lemma 4.3 in [4]):

$$\mathbb{P}(X = x) \propto \exp(f(x)) = \exp\left(\sum_{j=1}^m \rho(J_j \cdot x + g_j) + h^T x\right)$$

where $J_j$ is the $j$-th column of $J$ and $\rho(x) = \log(e^x + e^{-x})$. From this, it can be shown that the marginal distribution is an MRF of order at most $d$.

We now define $s$, the maximum number of latent variables connected to the MRF neighborhood of an observed variable:

$$s := \max_{u \in [n]} \sum_{j=1}^m \mathbb{1}\{\exists i \in [n] \setminus \{u\} \text{ and } S \subseteq [n] \text{ s.t. } u, i \in S \text{ and } \hat{f}(S) \neq 0 \text{ and } J_{i,j} \neq 0\}.$$

The MRF neighborhood of an observed variable is a subset of the two-hop neighborhood of the observed variable in the RBM; typically the two neighborhoods are identical. Therefore, an upper bound on $s$ is obtained as the maximum number of latent variables connected to the two-hop neighborhood of an observed variable in the RBM.

Finally, we define a proxy to the conditional mutual information, which is used extensively in our analysis. For random variables $X_u \in \{-1, 1\}$, $X_I \in \{-1, 1\}^{|I|}$, and $X_S \in \{-1, 1\}^{|S|}$, let

$$\nu_{u,I|S} := \mathbb{E}_{R,G}\left[\mathbb{E}_{X_S}\left[|\mathbb{P}(X_u = R, X_I = G|X_S) - \mathbb{P}(X_u = R|X_S)\mathbb{P}(X_I = G|X_S)|\right]\right]$$

where $R$ and $G$ come from uniform distributions over $\{-1, 1\}$ and $\{-1, 1\}^{|I|}$, respectively. This quantity forms a lower bound on the conditional mutual information (e.g., see Lemma 2.5 in [10]):

$$\sqrt{\frac{1}{2} I(X_u; X_I | X_S)} \geq \nu_{u,I|S}.$$

We also define an empirical version of this proxy, with the probabilities and the expectation over $X_S$ replaced by their averages from samples:

$$\hat{\nu}_{u,I|S} := \mathbb{E}_{R,G}\left[\hat{\mathbb{E}}_{X_S}\left[\left|\hat{\mathbb{P}}(X_u = R, X_I = G|X_S) - \hat{\mathbb{P}}(X_u = R|X_S)\hat{\mathbb{P}}(X_I = G|X_S)\right|\right]\right].$$

# 3    Learning Restricted Boltzmann Machines with sparse latent variables

To find the MRF neighborhood of an observed variable $u$ (i.e., observed variable $X_u$; we use the index and the variable interchangeably when no confusion is possible), our algorithm takes the following steps, similar to those of the algorithm of [10]:

1. Fix parameters $s$, $\tau'$, $L$. Fix observed variable $u$. Set $S := \emptyset$.
2. While $|S| \leq L$ and there exists a set of observed variables $I \subseteq [n] \setminus \{u\} \setminus S$ of size at most $2^s$ such that $\hat{\nu}_{u,I|S} > \tau'$, set $S := S \cup I$.
3. For each $i \in S$, if $\hat{\nu}_{u,i|S \setminus \{i\}} < \tau'$, remove $i$ from $S$.
4. Return set $S$ as an estimate of the neighborhood of $u$.

We use

$$L = 8/(\tau')^2, \quad \tau' = \frac{1}{(4d)^{2^s}} \left(\frac{1}{d}\right)^{2^s(2^s+1)} \tau, \text{ and } \tau = \frac{1}{2}\frac{4\alpha^2(e^{-2\gamma})^{d+D-1}}{d^{4d}2^{d+1}\binom{D}{d-1}\gamma e^{2\gamma}},$$

where $\tau$ is exactly as in [10] when adapted to the RBM setting. In the above, $d$ is a property of the RBM, and $D$, $\alpha$, and $\gamma$ are properties of the MRF of the observed variables.

With high probability, Step 2 is guaranteed to add to $S$ all the MRF neighbors of $u$, and Step 3 is guaranteed to prune from $S$ any non-neighbors of $u$. Therefore, with high probability, in Step 4 $S$ is exactly the MRF neighborhood of $u$. In the original algorithm of [10], the guarantees of Step 2 were based on this result: if $S$ does not contain the entire neighborhood of $u$, then $\nu_{u,I|S} \geq 2\tau$ for some set $I$ of size at most $d-1$. As a consequence, Step 2 entailed a search over size $d-1$ sets. The analogous result in our setting is given in Theorem 5, which guarantees the existence of a set $I$ of size at most $2^s$, thus reducing the search to sets of this size. This theorem follows immediately from Theorem 4, the key structural result of our paper.

**Theorem 4.** *Fix observed variable $u$ and subsets of observed variables $I$ and $S$, such that all three are disjoint. Suppose that $I$ is a subset of the MRF neighborhood of $u$ and that $|I| \leq d-1$. Then there exists a subset $I' \subseteq I$ with $|I'| \leq 2^s$ such that*

$$\nu_{u,I'|S} \geq \frac{1}{(4d)^{2^s}}\left(\frac{1}{d}\right)^{2^s(2^s+1)} \nu_{u,I|S}.$$

Using the result in Theorem 4, we now state and prove Theorem 5.

**Theorem 5.** *Fix an observed variable $u$ and a subset of observed variables $S$, such that the two are disjoint. Suppose that $S$ does not contain the entire MRF neighborhood of $u$. Then there exists some subset $I$ of the MRF neighborhood of $u$ with $|I| \leq 2^s$ such that*

$$\nu_{u,I|S} \geq \frac{1}{(4d)^{2^s}}\left(\frac{1}{d}\right)^{2^s(2^s+1)}\frac{4\alpha^2(e^{-2\gamma})^{d+D-1}}{d^{4d}2^{d+1}\binom{D}{d-1}\gamma e^{2\gamma}} = 2\tau'.$$

*Proof.* By Theorem 4.6 in [10], we have that there exists some subset $I$ of neighbors of $u$ with $|I| \leq d-1$ such that

$$\nu_{u,I|S} \geq \frac{4\alpha^2(e^{-2\gamma})^{d+D-1}}{d^{4d}2^{d+1}\binom{D}{d-1}\gamma e^{2\gamma}} = 2\tau.$$

Then, by Theorem 4, we have that there exists some subset $I' \subseteq I$ with $|I'| \leq 2^s$ such that

$$\nu_{u,I'|S} \geq \frac{1}{(4d)^{2^s}}\left(\frac{1}{d}\right)^{2^s(2^s+1)} 2\tau = \frac{1}{(4d)^{2^s}}\left(\frac{1}{d}\right)^{2^s(2^s+1)}\frac{4\alpha^2(e^{-2\gamma})^{d+D-1}}{d^{4d}2^{d+1}\binom{D}{d-1}\gamma e^{2\gamma}} = 2\tau'.$$

$\square$

Theorem 6 states the guarantees of our algorithm. The analysis is very similar to that in [10], and is deferred to the Appendix B. Then, Section 4 sketches the proof of Theorem 4.

**Theorem 6.** *Fix $\omega > 0$. Suppose we are given $M$ samples from an RBM, where*

$$M \geq \frac{60 \cdot 2^{2L}}{(\tau')^2 (e^{-2\gamma})^{2L}} \left( \log(1/\omega) + \log(L + 2^s + 1) + (L + 2^s + 1)\log(2n) + \log 2 \right).$$

*Then with probability at least $1 - \omega$, our algorithm, when run from each observed variable $u$, recovers the correct neighborhood of $u$. Each run of the algorithm takes $O(MLn^{2^s+1})$ time.*

## 4 Proof sketch of structural result

The proofs of the lemmas in this section can be found in the Appendix A. Consider the mutual information proxy when the values of $X_u$, $X_I$, and $X_S$ are fixed:

$$\nu_{u,I|S}(x_u, x_I | x_S)$$
$$= \left| \mathbb{P}(X_u = x_u, X_I = x_I | X_S = x_S) - \mathbb{P}(X_u = x_u | X_S = x_S)\mathbb{P}(X_I = x_I | X_S = x_S) \right|.$$

We first establish a version of Theorem 4 for $\nu_{u,I|S}(x_u, x_I | x_S)$, and then generalize it to $\nu_{u,I|S}$.

In Lemma 7, we express $\nu_{u,I|S}(x_u, x_I | x_S)$ as a sum over configurations of latent variables $U$ connected to observed variables in $I$. Note that $|U| \leq s$, so the summation is over at most $2^s$ terms.

**Lemma 7.** *Fix observed variable $u$ and subsets of observed variables $I$ and $S$, such that all three are disjoint. Suppose that $I$ is a subset of the MRF neighborhood of $u$. Then*

$$\nu_{u,I|S}(x_u, x_I | x_S) = \left| \sum_{q_U \in \{-1,1\}^{|U|}} \left( \sum_{q_{\sim U} \in \{-1,1\}^{m-|U|}} \bar{f}(q, x_u, x_S) \right) \prod_{i \in I} \sigma(2x_i(J^{(i)} \cdot q + h_i)) \right|$$

*for some function $\bar{f}$, where $U$ is the set of latent variables connected to observed variables in $I$, $J^{(i)}$ is the $i$-th row of $J$, and the entries of $q_{\sim U}$ in the expression $J^{(i)} \cdot q$ are arbitrary.*

Lemma 8 gives a generic non-cancellation result for expressions of the form $\left| \sum_{i=1}^m a_i \prod_{j=1}^n x_{i,j} \right|$. Then, Lemma 9 applies this result to the form of $\nu_{u,I|S}(x_u, x_I | x_S)$ in Lemma 7, and guarantees the existence of a subset $I' \subseteq I$ with $|I'| \leq 2^s$ such that $\nu_{u,I'|S}(x_u, x_{I'} | x_S)$ is within a bounded factor of $\nu_{u,I|S}(x_u, x_I | x_S)$.

**Lemma 8.** *Let $x_{1,1}, ..., x_{m,n} \in [-1, 1]$, with $n > m$. Then, for any $a \in \mathbb{R}^m$, there exists a subset $S \subseteq [n]$ with $|S| \leq m$ such that*

$$\left| \sum_{i=1}^m a_i \prod_{j \in S} x_{i,j} \right| \geq \frac{1}{4^m} \left( \frac{1}{n} \right)^{m(m+1)} \left| \sum_{i=1}^m a_i \prod_{j=1}^n x_{i,j} \right|.$$

We remark that, in this general form, Lemma 8 is optimal in the size of the subset that it guarantees not to be cancelled. That is, there are examples with $\sum_{i=1}^m a_i \prod_{j=1}^n x_{i,j} \neq 0$ but $\sum_{i=1}^m a_i \prod_{j \in S} x_{i,j} = 0$ for all subsets $S \subseteq [n]$ with $|S| \leq m - 1$. See the Appendix A for a more detailed discussion.

**Lemma 9.** *Fix observed variable $u$ and subsets of observed variables $I$ and $S$, such that all three are disjoint. Suppose that $I$ is a subset of the MRF neighborhood of $u$. Fix any assignments $x_u$, $x_I$, and $x_S$. Then there exists a subset $I' \subseteq I$ with $|I'| \leq 2^s$ such that*

$$\nu_{u,I'|S}(x_u, x_{I'} | x_S) \geq \frac{1}{4^{2^s}} \left( \frac{1}{|I|} \right)^{2^s(2^s+1)} \nu_{u,I|S}(x_u, x_I | x_S)$$

*where $x_{I'}$ agrees with $x_I$.*

Finally, Lemma 10 extends the result about $\nu_{u,I|S}(x_u, x_I | x_S)$ to a result about $\nu_{u,I|S}$. The difficulty lies in the fact that the subset $I'$ guaranteed to exist in Lemma 9 may be different for different configurations $(x_u, x_I, x_S)$. Nevertheless, the number of subsets $I'$ with $|I'| \leq 2^s$ is smaller than the number of configurations $(x_u, x_I, x_S)$, so we obtain a viable bound via the pigeonhole principle.

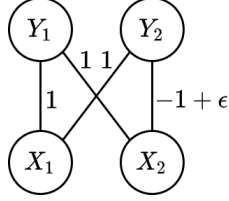

Figure 2: RBM with $\alpha \to 0$ as $\epsilon \to 0$ and $\alpha^* = 1$, $\beta^* = 2$ when $0 \le \epsilon \le 2$. The $X$ variables represent observed variables, the $Y$ variables represent latent variables, and the edges represent non-zero interactions between variables. All external field terms are zero.

**Lemma 10.** *Fix observed variable $u$ and subsets of observed variables $I$ and $S$, such that all three are disjoint. Suppose that $I$ is a subset of the MRF neighborhood of $u$. Then there exists a subset $I' \subseteq I$ with $|I'| \le 2^s$ such that*

$$\nu_{u,I'|S} \ge \frac{1}{(4|I|)^{2^s}} \left( \frac{1}{|I|} \right)^{2^s(2^s+1)} \nu_{u,I|S}.$$

This result completes the proof of Theorem 4.

## 5  Making good predictions independently of the minimum potential

Figure 2 shows an RBM for which $\alpha$ can be arbitrarily small, while $\alpha^* = 1$ and $\beta^* = 2$. That is, the induced MRF can be degenerate, while the RBM itself has interactions that are far from degenerate. This is problematic: the sample complexity of our algorithm, which scales with the inverse of $\alpha$, can be arbitrarily large, even for seemingly well-behaved RBMs. In particular, we note that $\alpha$ is an opaque property of the RBM, and it is *a priori* unclear how small it is.

We emphasize that this scaling with the inverse of $\alpha$ is necessary information-theoretically [22]. All current algorithms for learning MRFs and RBMs have this dependency, and it is impossible to remove it while still guaranteeing the recovery of the structure of the model.

Instead, in this section we give an algorithm that learns an RBM with small prediction error, independently of $\alpha$. We necessarily lose the guarantee on structure recovery, but we guarantee accurate prediction even for RBMs in which $\alpha$ is arbitrarily degenerate. The algorithm is composed of a structure learning step that recovers the MRF neighborhoods corresponding to large potentials, and a regression step that estimates the values of these potentials.

### 5.1  Structure learning algorithm

The structure learning algorithm is guaranteed to recover the MRF neighborhoods corresponding to potentials that are at least $\zeta$ in absolute value. The guarantees of the algorithm are stated in Theorem 11, which is proved in the Appendix D.

The main differences between this algorithm and the one in Section 3 are: first, the thresholds for $\hat{\nu}_{u,I|S}$ are defined in terms of $\zeta$ instead of $\alpha$, and second, the threshold for $\hat{\nu}_{u,I|S}$ in the additive step (Step 2) is smaller than that used in the pruning step (Step 3), in order to guarantee the pruning of all non-neighbors. The algorithm is described in detail in the Appendix C.

**Theorem 11.** *Fix $\omega > 0$. Suppose we are given $M$ samples from an RBM, where $M$ is as in Theorem 6 if $\alpha$ were equal to*

$$\alpha = \frac{\zeta}{\sqrt{3} \cdot 2^{D/2+2^s} \cdot D^{2^{s-1}(2^s+2)}}.$$

*Then with probability at least $1 - \omega$, our algorithm, when run starting from each observed variable $u$, recovers a subset of the MRF neighbors of $u$, such that all neighbors which are connected to $u$ through a Fourier coefficient of absolute value at least $\zeta$ are included in the subset. Each run of the algorithm takes $O(MLn^{2^s+1})$ time.*

## 5.2 Regression algorithm

Note that

$$\mathbb{P}(X_u = 1 | X_{[n]\setminus\{u\}} = x_{[n]\setminus\{u\}}) = \sigma\left(2 \sum_{S \subseteq [n]\setminus\{u\}} \hat{f}(S \cup \{u\})\chi_S(x)\right).$$

Therefore, following the approach of [26], we can frame the recovery of the Fourier coefficients as a regression task. Let $n(u)$ be the set of MRF neighbors of $u$ recovered by the algorithm in Section 5.1. Note that $|n(u)| \leq D$. Let $z \in \{-1,1\}^{2^{|n(u)|}}$, $w \in \mathbb{R}^{2^{|n(u)|}}$, and $y \in \{-1,1\}$, with $z_S = \chi_S(X)$, $w_S = 2\hat{f}(S \cup \{u\})$, and $y = X_u$, for all subsets $S \subseteq n(u)$. Then, if $n(u)$ were equal to the true set of MRF neighbors, we could rewrite the conditional probability statement above as

$$\mathbb{P}(y = 1|z) = \sigma(w \cdot z), \quad \text{with } ||w||_1 \leq 2\gamma.$$

Then, finding an estimate $\hat{w}$ would amount to a constrained regression problem. In our setting, we solve the same problem, and we show that the resulting estimate has small prediction error. We estimate $\hat{w}$ as follows:

$$\hat{w} \in \underset{w \in \mathbb{R}^{|n(u)|}}{\operatorname{argmin}} \frac{1}{M} \sum_{i=1}^{M} l(y^{(i)}(w \cdot z^{(i)})) \quad \text{s.t. } ||w||_1 \leq 2\gamma,$$

where we assume we have access to $M$ i.i.d. samples $(z, y)$, and where $l : \mathbb{R} \to \mathbb{R}$ is the loss function

$$l(y(w \cdot z)) = \ln(1 + e^{-y(w\cdot z)}) = \begin{cases} -\ln\sigma(w \cdot z), & \text{if } y = 1 \\ -\ln(1 - \sigma(w \cdot z)), & \text{if } y = -1 \end{cases}.$$

The objective above is convex, and the problem is solvable in time $\tilde{O}((2^D)^4)$ by the $l_1$-regularized logistic regression method described in [17]. Then, Theorem 12 gives theoretical guarantees for the prediction error achieved by this regression algorithm. The proof is deferred to the Appendix D.

**Theorem 12.** *Fix $\delta > 0$ and $\epsilon > 0$. Suppose that we are given neighborhoods $n(u)$ satisfying the guarantees of Theorem 11 for each observed variable $u$. Suppose that we are given $M$ samples from the RBM, and that we have*

$$M = \Omega\left(\gamma^2 \ln(8 \cdot n \cdot 2^D/\delta)/\epsilon^2\right), \quad \zeta \leq \frac{\sqrt{\epsilon}}{D^d\sqrt{1 + e^{2\gamma}}}.$$

*Let $z_u$ and $\hat{w}_u$ be the features and the estimate of the weights when the regression algorithm is run at observed variable $u$. Then, with probability at least $1 - \delta$, for all variables $u$,*

$$\mathbb{E}\left[\left(\mathbb{P}(X_u = 1|X_{\setminus u}) - \sigma(\hat{w}_u \cdot z_u)\right)^2\right] \leq \epsilon.$$

The sample complexity of the combination of structure learning and regression is given by the sum of the sample complexities of the two algorithms. When $\delta$ is constant, the number of samples required by regression is absorbed by the number of samples required by strucutre learning. For structure learning, plugging in the upper bound on $\zeta$ required by Theorem 12, we get that the sample complexity is exponential in $\epsilon^{-1}$. Note that the factors $D^d$ and $\sqrt{1 + e^{2\gamma}}$ in the upper bound on $\zeta$, as well as the factors that appear in Theorem 11 from the relative scaling of $\alpha$ and $\zeta$, do not influence the sample complexity much, because factors of similar order already appear in the sample complexity of the structure learning algorithm. Overall, for constant $\delta$ and constant $\epsilon$, the combined sample complexity is comparable to that of the algorithm in Section 3, without the $\alpha$ dependency.

## Broader impact

This work does not present any foreseeable societal consequence.

## Funding disclosure

G.B. was supported in part by MIT-IBM Watson AI Lab and NSF CAREER award CCF-1940205.

## Footnotes

[2]This definition holds if each assignment of the random variables has positive probability, which is satisfied by the models considered in this paper.

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
