[Supplementary Material]

# A Proof of Theorem 4

## A.1 Proof of Lemma 7

We first state and prove Lemmas 13, 14, and 15, which provide the foundation for the proof of Lemma 7.

**Lemma 13.** *Let $f(x) = \sum_{j=1}^{m} \rho(J_j \cdot x + g_j) + h^T x$, where $x \in \{-1, 1\}^n$, $J \in \mathbb{R}^{n \times m}$, $h \in \mathbb{R}^n$, and $g \in \mathbb{R}^m$. Then*

$$
\begin{aligned}
&\mathbb{E}_{\mathcal{U}}[\mathbb{1}_{X_I = x_I} e^{f(x)} | X_S = x_S] \\
&= \sum_{q \in \{-1,1\}^m} e^{g \cdot q} \prod_{i \in S} e^{x_i (J^{(i)} \cdot q + h_i)} \prod_{i \in [n] \setminus S} \cosh(J^{(i)} \cdot q + h_i) \prod_{i \in I} \sigma(2 x_i (J^{(i)} \cdot q + h_i))
\end{aligned}
$$

*where $\mathcal{U}$ denotes the uniform distribution over $\{-1, 1\}^n$ and where $J^{(i)}$ denotes the $i$-th row of $J$.*

*Proof.*

$$
\begin{aligned}
&\mathbb{E}_{\mathcal{U}}[\mathbb{1}_{X_I = x_I} e^{f(x)} | X_S = x_S] \\
&= \frac{1}{2^{n-|S|}} \sum_{x \in \{-1,1\}^n} \mathbb{1}_{X_S = x_S, X_I = x_I} \cdot e^{h \cdot x} \prod_{j=1}^{m} \left( e^{J_j \cdot x + g_j} + e^{-J_j \cdot x - g_j} \right) \\
&= \frac{1}{2^{n-|S|}} \sum_{x \in \{-1,1\}^n} \mathbb{1}_{X_S = x_S, X_I = x_I} \cdot e^{h \cdot x} \sum_{q \in \{-1,1\}^m} e^{(x^T J + g^T) q} \\
&= \frac{1}{2^{n-|S|}} \sum_{x \in \{-1,1\}^n} \mathbb{1}_{X_S = x_S, X_I = x_I} \sum_{q \in \{-1,1\}^m} e^{x^T (Jq + h)} e^{g \cdot q} \\
&= \frac{1}{2^{n-|S|}} \sum_{q \in \{-1,1\}^m} \sum_{x \in \{-1,1\}^n} \mathbb{1}_{X_S = x_S, X_I = x_I} \cdot e^{x^T (Jq + h)} e^{g \cdot q} \\
&= \frac{1}{2^{n-|S|}} \sum_{q \in \{-1,1\}^m} e^{g \cdot q} \sum_{x \in \{-1,1\}^n} \mathbb{1}_{X_S = x_S, X_I = x_I} \cdot e^{\sum_{i=1}^{n} x_i (J^{(i)} \cdot q + h_i)} \\
&= \frac{1}{2^{n-|S|}} \sum_{q \in \{-1,1\}^m} e^{g \cdot q} \left( \sum_{x_{[n] \setminus (S \cup I)}} \prod_{i \in [n] \setminus (S \cup I)} e^{x_i (J^{(i)} \cdot q + h_i)} \right) \prod_{i \in S \cup I} e^{x_i (J^{(i)} \cdot q + h_i)} \\
&= \frac{1}{2^{n-|S|}} \sum_{q \in \{-1,1\}^m} e^{g \cdot q} \prod_{i \in [n] \setminus (S \cup I)} \left( e^{J^{(i)} \cdot q + h_i} + e^{-J^{(i)} \cdot q - h_i} \right) \prod_{i \in S \cup I} e^{x_i (J^{(i)} \cdot q + h_i)} \\
&= \sum_{q \in \{-1,1\}^m} e^{g \cdot q} \prod_{i \in S} e^{x_i (J^{(i)} \cdot q + h_i)} \prod_{i \in [n] \setminus (S \cup I)} \cosh(J^{(i)} \cdot q + h_i) \prod_{i \in I} \frac{e^{x_i (J^{(i)} \cdot q + h_i)}}{2} \\
&= \sum_{q \in \{-1,1\}^m} e^{g \cdot q} \prod_{i \in S} e^{x_i (J^{(i)} \cdot q + h_i)} \prod_{i \in [n] \setminus S} \cosh(J^{(i)} \cdot q + h_i) \prod_{i \in I} \frac{e^{x_i (J^{(i)} \cdot q + h_i)}}{2 \cosh(J^{(i)} \cdot q + h_i)} \\
&= \sum_{q \in \{-1,1\}^m} e^{g \cdot q} \prod_{i \in S} e^{x_i (J^{(i)} \cdot q + h_i)} \prod_{i \in [n] \setminus S} \cosh(J^{(i)} \cdot q + h_i) \prod_{i \in I} \sigma(2 x_i (J^{(i)} \cdot q + h_i)).
\end{aligned}
$$

$\square$

**Lemma 14.** *Fix subsets of observed variables $I$ and $S$, such that the two are disjoint. Then*

$$
\mathbb{P}(X_I = x_I | X_S = x_S) = \sum_{q \in \{-1,1\}^m} \lambda(q, x_S) \prod_{i \in I} \sigma(2 x_i (J^{(i)} \cdot q + h_i))
$$

*where*

$$
\lambda(q, x_S) = \frac{e^{g \cdot q} \prod_{i \in S} e^{x_i (J^{(i)} \cdot q + h_i)} \prod_{i \in [n] \setminus S} \cosh(J^{(i)} \cdot q + h_i)}{\sum_{q' \in \{-1,1\}^m} e^{g \cdot q'} \prod_{i \in S} e^{x_i (J^{(i)} \cdot q' + h_i)} \prod_{i \in [n] \setminus S} \cosh(J^{(i)} \cdot q' + h_i)}.
$$

*Proof.* The MRF of the observed variables has a probability mass function that is proportional to $\exp(f(x))$, where $f(x)$ is as in Lemma 13. Then

$$\mathbb{P}(X_I = x_I | X_S = x_S)$$

$$= \frac{\mathbb{P}(X_S = x_S, X_I = x_I)}{\mathbb{P}(X_S = x_S)}$$

$$= \frac{\mathbb{E}[\mathbb{1}_{X_S = x_S, X_I = x_I}]}{\mathbb{E}[\mathbb{1}_{X_S = x_S}]}$$

$$= \frac{\frac{1}{Z} \sum_{x \in \{-1,1\}^n} \mathbb{1}_{X_S = x_S, X_I = x_I} \cdot e^{f(x)}}{\frac{1}{Z} \sum_{x \in \{-1,1\}^n} \mathbb{1}_{X_S = x_S} \cdot e^{f(x)}}$$

$$= \frac{\frac{1}{2^{n-|S|}} \sum_{x \in \{-1,1\}^n} \mathbb{1}_{X_S = x_S, X_I = x_I} \cdot e^{f(x)}}{\frac{1}{2^{n-|S|}} \sum_{x \in \{-1,1\}^n} \mathbb{1}_{X_S = x_S} \cdot e^{f(x)}}$$

$$= \frac{\mathbb{E}_{\mathcal{U}}[\mathbb{1}_{X_I = x_I} e^{f(x)} | X_S = x_S]}{\mathbb{E}_{\mathcal{U}}[e^{f(x)} | X_S = x_S]}$$

$$= \frac{\sum_{q \in \{-1,1\}^m} e^{g \cdot q} \prod_{i \in S} e^{x_i(J^{(i)} \cdot q + h_i)} \prod_{i \in [n] \setminus S} \cosh(J^{(i)} \cdot q + h_i) \prod_{i \in I} \sigma(2x_i(J^{(i)} \cdot q + h_i))}{\sum_{q' \in \{-1,1\}^m} e^{g \cdot q'} \prod_{i \in S} e^{x_i(J^{(i)} \cdot q' + h_i)} \prod_{i \in [n] \setminus S} \cosh(J^{(i)} \cdot q' + h_i)}$$

$$= \sum_{q \in \{-1,1\}^m} \lambda(q, x_S) \prod_{i \in I} \sigma(2x_i(J^{(i)} \cdot q + h_i)).$$

$\square$

**Lemma 15.** *Fix observed variable $u$ and subsets of observed variables $I$ and $S$, such that all three are disjoint. Then*

$$\nu_{u,I|S}(x_u, x_I | x_S) = \left| \sum_{q \in \{-1,1\}^m} \bar{f}(q, x_u, x_S) \prod_{i \in I} \sigma(2x_i(J^{(i)} \cdot q + h_i)) \right|$$

*where $J^{(i)}$ denotes the $i$-th row of $J$ and where*

$$\bar{f}(q, x_u, x_S) = \lambda(q, x_S) \left[ \sigma(2x_u(J^{(u)} \cdot q + h_u)) - \mathbb{E}_{q' \sim \lambda(\cdot, x_S)} \sigma(2x_u(J^{(u)} \cdot q' + h_u)) \right],$$

$$\lambda(q, x_S) = \frac{e^{g \cdot q} \prod_{i \in S} e^{x_i(J^{(i)} \cdot q + h_i)} \prod_{i \in [n] \setminus S} \cosh(J^{(i)} \cdot q + h_i)}{\sum_{q' \in \{-1,1\}^m} e^{g \cdot q'} \prod_{i \in S} e^{x_i(J^{(i)} \cdot q' + h_i)} \prod_{i \in [n] \setminus S} \cosh(J^{(i)} \cdot q' + h_i)}.$$

*Proof.* We apply Lemma 14 to the terms in the definition of $\nu_{u,I|S}(x_u, x_I | x_S)$:

$$\mathbb{P}(X_u = x_u, X_I = x_I | X_S = x_S) - \mathbb{P}(X_u = x_u | X_S = x_S)\mathbb{P}(X_I = x_I | X_S = x_S)$$

$$= \sum_{q \in \{-1,1\}^m} \lambda(q, x_S) \sigma(2x_u(J^{(u)} \cdot q + h_u)) \prod_{i \in I} \sigma(2x_i(J^{(i)} \cdot q + h_i))$$

$$- \left[ \sum_{q \in \{-1,1\}^m} \lambda(q, x_S) \sigma(2x_u(J^{(u)} \cdot q + h_u)) \right] \left[ \sum_{q \in \{-1,1\}^m} \lambda(q, x_S) \prod_{i \in I} \sigma(2x_i(J^{(i)} \cdot q + h_i)) \right]$$

$$= \sum_{q \in \{-1,1\}^m} \sum_{q' \in \{-1,1\}^m} \lambda(q, x_S) \lambda(q', x_S) \sigma(2x_u(J^{(u)} \cdot q + h_u)) \prod_{i \in I} \sigma(2x_i(J^{(i)} \cdot q + h_i))$$

$$- \sum_{q \in \{-1,1\}^m} \sum_{q' \in \{-1,1\}^m} \lambda(q, x_S) \lambda(q', x_S) \sigma(2x_u(J^{(u)} \cdot q' + h_u)) \prod_{i \in I} \sigma(2x_i(J^{(i)} \cdot q + h_i))$$

$$= \sum_{q \in \{-1,1\}^m} \lambda(q, x_S) \left[ \sigma(2x_u(J^{(u)} \cdot q + h_u)) - \mathbb{E}_{q' \sim \lambda(\cdot, x_S)} \sigma(2x_u(J^{(u)} \cdot q' + h_u)) \right]$$

$$\cdot \prod_{i \in I} \sigma(2x_i(J^{(i)} \cdot q + h_i))$$

$$= \sum_{q \in \{-1,1\}^m} \bar{f}(q, x_u, x_S) \prod_{i \in I} \sigma(2x_i(J^{(i)} \cdot q + h_i)).$$

□

*Proof of Lemma 7.* Note that, if $J_{i,j} = 0$ for all $i \in I$, then the term $\prod_{i \in I} \sigma(2x_i(J^{(i)} \cdot q + h_i))$ is independent of the value of $q_j$. Let $U = \{j \in [m] : J_{i,j} \neq 0 \text{ for some } i \in I\}$ be the set of latent variables with connections to observed variables in $I$. By Lemma 15, we can write then

$$\nu_{u,I|S}(x_u, x_I | x_S) = \left| \sum_{q_U \in \{-1,1\}^{|U|}} \left( \sum_{q \sim U \in \{-1,1\}^{m-|U|}} \bar{f}(q, x_u, x_S) \right) \prod_{i \in I} \sigma(2x_i(J^{(i)} \cdot q + h_i)) \right|.$$

□

## A.2  Proof of Lemma 8

A special case of Lemma 8 is given in Lemma 16. Then, we prove Lemma 8. Lastly, Section A.2.1 shows that these lemmas are tight in the size of the subset that they guarantee not to be cancelled.

**Lemma 16.** *Let $x_{1,1}, ..., x_{m,m+1} \in [-1, 1]$. Then, for any $a \in \mathbb{R}^m$, there exists a subset $S \subseteq [m+1]$ with $|S| \leq m$ such that*

$$\left| \sum_{i=1}^m a_i \prod_{j \in S} x_{i,j} \right| \geq \frac{1}{2^m - 1} \cdot \left| \sum_{i=1}^m a_i \prod_{j=1}^{m+1} x_{i,j} \right|.$$

*Proof.* We prove the claim by induction on $m$.

**Base case:** For $m = 1$, we have

$$|ax_{1,1}| = \frac{|ax_{1,1}x_{1,2}|}{|x_{1,2}|} \geq |ax_{1,1}x_{1,2}|$$

$$|ax_{1,2}| = \frac{|ax_{1,1}x_{1,2}|}{|x_{1,1}|} \geq |ax_{1,1}x_{1,2}|$$

Therefore, the claim holds, with either $S = \{1\}$ or $S = \{2\}$. Note that if any of $a$, $x_{1,1}$, or $x_{1,2}$ is zero, then $ax_{1,1}x_{1,2} = 0$ and the claim holds trivially.

**Induction step:** Assume the claim holds for $m - 1$.

Suppose $|\sum_{i=1}^m a_i \prod_{j \in S} x_{i,j}| < \frac{1}{2^m - 1} \cdot \left| \sum_{i=1}^m a_i \prod_{j=1}^{m+1} x_{i,j} \right|$ for any $S \subseteq [m]$; otherwise the induction step follows. By the triangle inequality, we have

$$\left| \sum_{i=1}^m a_i \prod_{j=1}^{m+1} x_{i,j} \right| = \left| \sum_{i=1}^m a_i x_{i,m+1} \prod_{j=1}^m x_{i,j} \right|$$

$$\leq \left| \sum_{i=1}^m a_i(x_{i,m+1} - x_{m,m+1}) \prod_{j=1}^m x_{i,j} \right| + |x_{m,m+1}| \cdot \left| \sum_{i=1}^m a_i \prod_{j=1}^m x_{i,j} \right|.$$

For the first term on the right-hand side, we clearly have $x_{i,m+1} - x_{m,m+1} = 0$ at $i = m$. Therefore, the term is of the form $\sum_{i=1}^{m-1} b_i \sum_{j=1}^m y_{i,j}$, so we can apply the inductive claim for $m - 1$. Therefore, there exists a subset $S^* \subseteq [m]$ with $|S^*| \leq m - 1$ such that

$$\left| \sum_{i=1}^m a_i(x_{i,m+1} - x_{m,m+1}) \prod_{j \in S^*} x_{i,j} \right| \geq \frac{1}{2^{m-1} - 1} \cdot \left| \sum_{i=1}^m a_i(x_{i,m+1} - x_{m,m+1}) \prod_{j=1}^m x_{i,j} \right|.$$

Overall, we get then the inequality:

$$\left| \sum_{i=1}^m a_i \prod_{j=1}^{m+1} x_{i,j} \right|$$

$$\leq (2^{m-1} - 1) \cdot \left| \sum_{i=1}^{m} a_i(x_{i,m+1} - x_{m,m+1}) \prod_{j \in S^*} x_{i,j} \right| + |x_{m,m+1}| \cdot \left| \sum_{i=1}^{m} a_i \prod_{j=1}^{m} x_{i,j} \right|$$

$$\leq (2^{m-1} - 1) \cdot \left| \sum_{i=1}^{m} a_i x_{i,m+1} \prod_{j \in S^*} x_{i,j} \right| + (2^{m-1} - 1) \cdot |x_{m,m+1}| \cdot \left| \sum_{i=1}^{m} a_i \prod_{j \in S^*} x_{i,j} \right|$$

$$+ |x_{m,m+1}| \cdot \left| \sum_{i=1}^{m} a_i \prod_{j=1}^{m} x_{i,j} \right|$$

$$\leq (2^{m-1} - 1) \cdot \left| \sum_{i=1}^{m} a_i x_{i,m+1} \prod_{j \in S^*} x_{i,j} \right| + \left( \frac{2^{m-1} - 1}{2^m - 1} + \frac{1}{2^m - 1} \right) \cdot \left| \sum_{i=1}^{m} a_i \prod_{j=1}^{m+1} x_{i,j} \right|$$

where in the last inequality we used that $|x_{m,m+1}| \leq 1$ and our supposition that for all $S \subseteq [m]$, $|\sum_{i=1}^{m} a_i \prod_{j \in S} x_{i,j}| < \frac{1}{2^m - 1} \cdot \left| \sum_{i=1}^{m} a_i \prod_{j=1}^{m+1} x_{i,j} \right|$. Then, reordering:

$$\left| \sum_{i=1}^{m} a_i x_{i,m+1} \prod_{j \in S^*} x_{i,j} \right| \geq \frac{1 - \frac{2^{m-1}-1}{2^m-1} - \frac{1}{2^m-1}}{2^{m-1} - 1} \left| \sum_{i=1}^{m} a_i \prod_{j=1}^{m+1} x_{i,j} \right| = \frac{1}{2^m - 1} \left| \sum_{i=1}^{m} a_i \prod_{j=1}^{m+1} x_{i,j} \right|.$$

Then, in this case, $S^* \cup \{m+1\}$ is the desired subset. Note that we selected $S^*$ such that $|S^*| \leq m-1$, so $|S^* \cup \{m+1\}| \leq m$. $\qquad\square$

*Proof of Lemma 8.* Partition $[n]$ into $m+1$ subsets $Q_1 = [1, \lceil \frac{n}{m+1} \rceil], Q_2 = [\lceil \frac{n}{m+1} \rceil + 1, 2\lceil \frac{n}{m+1} \rceil], ..., Q_{m+1} = [m\lceil \frac{n}{m+1} \rceil + 1, n]$. Then, apply Lemma 16 to

$$\left| \sum_{i=1}^{m} a_i \prod_{j=1}^{m+1} \left( \prod_{k \in Q_j} x_{i,k} \right) \right|$$

where we know that $\prod_{k \in Q_j} x_{i,k} \in [-1, 1]$, for all $j$. Then, there exists a subset $S \subseteq [m+1]$ with $|S| \leq m$ such that

$$\left| \sum_{i=1}^{m} a_i \prod_{j \in S} \left( \prod_{k \in Q_j} x_{i,k} \right) \right| \geq \frac{1}{2^m - 1} \left| \sum_{i=1}^{m} a_i \prod_{j=1}^{m+1} \left( \prod_{k \in Q_j} x_{i,k} \right) \right|.$$

Let $S' = \bigcup_{j \in S} Q_j$. Then $S' \subseteq [n]$ with $|S'| \leq n - \lfloor \frac{n}{m+1} \rfloor \leq n - \frac{n}{m+1} + 1$, and

$$\left| \sum_{i=1}^{m} a_i \prod_{j \in S'} x_{i,j} \right| \geq \frac{1}{2^m - 1} \left| \sum_{i=1}^{m} a_i \prod_{j=1}^{n} x_{i,j} \right|.$$

Now, if $|S'| > m$, apply the same technique recursively to $S'$: partition it into $m+1$ equal subsets and apply Lemma 16. Continue until you obtain a subset of size at most $m$.

We now bound the number of iterations required. Let $n_t$ be the size of the set at timestep $t$ (at the beginning, $n_0 = n$). We have

$$n_t \leq n_{t-1} \left( 1 - \frac{1}{m+1} \right) + 1$$

$$\leq n_{t-2} \left( 1 - \frac{1}{m+1} \right)^2 + \left( 1 - \frac{1}{m+1} \right) + 1$$

$$\leq ...$$

$$\leq n \left( 1 - \frac{1}{m+1} \right)^t + \sum_{q=0}^{t-1} \left( 1 - \frac{1}{m+1} \right)^q$$

$$\leq n\left(1 - \frac{1}{m+1}\right)^t + m + 1.$$

Let $T$ be the smallest timestep such that $n_T < m + 2$. An upper bound on $T$ is obtained as

$$n\left(1 - \frac{1}{m+1}\right)^T < 1 \implies ne^{-2T/(m+1)} < 1$$

$$\implies T > \frac{m+1}{2}\ln(n)$$

where we used that $e^{-2x} \leq 1 - x$ for $0 \leq x \leq 1/2$. Because $T$ is an integer, the correct upper bound is $\frac{m+1}{2}\ln(n) + 1$. Then, at this step, we are guaranteed that $n_T \leq m + 1$. One more step may be required to go from size $m + 1$ to size $m$. Therefore, an upper bound on the number of steps until $n_t \leq m$ is $\frac{m+1}{2}\ln(n) + 2$. Then, the factor due to applications of Lemma 16 is

$$\left(\frac{1}{2^m - 1}\right)^{\frac{m+1}{2}\ln(n)+2} \geq \left(\frac{1}{2^m}\right)^{\frac{m+1}{2}\ln(n)+2} = \frac{1}{2^{m(m+1)/2\ln(n)+2m}}$$

$$= \frac{1}{4^m}\left(\frac{1}{n}\right)^{m(m+1)\log_2(e)/2} \geq \frac{1}{4^m}\left(\frac{1}{n}\right)^{m(m+1)}.$$

$\square$

### A.2.1 Tightness of non-cancellation result

Lemma 17 shows that, in the setting of Lemmas 16 and 8, it is possible for all subsets of size strictly less than $m$ to be completely cancelled. Therefore, the guarantee on the existence of a subset of size at most $m$ that is non-cancelled is tight.

We emphasize that, for the RBM setting, this result does not imply an impossibility of finding subsets of size less than $2^s$ with non-zero mutual information proxy. One reason for this is that, in the RBM setting, the terms of the sums that we are interested in have additional constraints which are not captured by the general setting of this section.

**Lemma 17.** *For any $c \in \mathbb{R}$, there exists some $x_{1,1}, ..., x_{m,m} \in [-1, 1]$ and some $a \in \mathbb{R}^m$ such that*

$$\left|\sum_{i=1}^m a_i \prod_{j=1}^m x_{i,j}\right| = c$$

*and for any subset $S \subseteq [m]$ with $|S| \leq m - 1$*

$$\left|\sum_{i=1}^m a_i \prod_{j \in S} x_{i,j}\right| = 0.$$

*Proof.* Let $x_{1,1} = ... = x_{1,m} = x_1, ..., x_{m,1} = ... = x_{m,m} = x_m$. Then we want to select some $x_1, ..., x_m \in [-1, 1]$ and some $a \in \mathbb{R}^m$ such that

$$\begin{bmatrix} x_1 & x_2 & \cdots & x_m \\ \vdots & \vdots & \ddots & \vdots \\ x_1^{m-1} & x_2^{m-1} & \cdots & x_m^{m-1} \\ x_1^m & x_2^m & \cdots & x_m^m \end{bmatrix}\begin{bmatrix} a_1 \\ \vdots \\ a_{m-1} \\ a_m \end{bmatrix} = \begin{bmatrix} 0 \\ \vdots \\ 0 \\ c \end{bmatrix}.$$

Select some arbitrary $x_1, ..., x_m \in [-1, 1]$ such that the matrix on the left-hand-side has full rank. Note that $(x, ..., x^{m-1}, x^m)$ for $x \in \mathbb{R}$ is a point on the moment curve, and it is known that any such $m$ distinct non-zero points are linearly independent. Therefore, any distinct non-zero $x_1, ..., x_m \in [-1, 1]$ will do. Then, by matrix inversion, there exists some $a \in \mathbb{R}^m$ such that the relation holds. $\square$

## A.3 Proof of Lemma 9

*Proof of Lemma 9.* Apply Lemma 8 to the form of $\nu_{u,I|S}(x_u, x_I|x_S)$ in Lemma 7, with

$$\sum_{q\sim U\in\{-1,1\}^{m-|U|}} \bar{f}(q, x_u, x_S)$$

treated as a coefficient (i.e., $a$ in Lemma 8) and

$$\sigma(2x_i(J^{(i)} \cdot q + h_i))$$

treated as a variable in $[-1, 1]$ (i.e., $x$ in Lemma 8). Then there exists a subset $I' \subseteq I$ with $|I'| \leq 2^{|U|}$ such that

$$\left| \sum_{q_U\in\{-1,1\}^{|U|}} \left( \sum_{q\sim U\in\{-1,1\}^{m-|U|}} \bar{f}(q, x_u, x_S) \right) \prod_{i\in I'} \sigma(2x_i(J^{(i)} \cdot q + h_i)) \right|$$

$$\geq \frac{1}{4^{2^{|U|}}} \left( \frac{1}{|I|} \right)^{2^{|U|}(2^{|U|}+1)} \nu_{u,I|S}(x_u, x_I|x_S).$$

Note that the latent variables connected to observed variables in $I'$ are a subset of $U$. Then, by Lemma 7, the expression on the left-hand side is equal to $\nu_{u,I'|S}(x_u, x_{I'}|x_S)$ where $x_{I'}$ agrees with $x_I$. Finally, note that $|U| \leq s$. $\qquad\square$

## A.4 Proof of Lemma 10

*Proof of Lemma 10.* We have that

$$\nu_{u,I|S} = \sum_{x_u,x_I,x_S} \frac{\mathbb{P}(X_S = x_S)}{2^{|I|+1}} \nu_{u,I|S}(x_u, x_I|x_S).$$

Hence, $\nu_{u,I|S}$ is a sum of $2^{|S|+|I|+1}$ terms $\nu_{u,I|S}(x_u, x_I|x_S)$. Lemma 9 applies to each term $\nu_{u,I|S}(x_u, x_I|x_S)$ individually. However, the subset $I'$ with $|I'| \leq 2^s$ that is guaranteed to exist by Lemma 9 may be a function of the specific assignment $x_u$, $x_I$, and $x_S$. Let $I^*(x_u, x_I|x_S)$ be the subset $I'$ with $|I'| \leq 2^s$ that is guaranteed to exist by Lemma 9 for assignment $x_u$, $x_I$, and $x_S$.

The number of non-empty subsets $I' \subseteq I$ with $|I'| \leq 2^s$ is at most $|I|^{2^s}$. Then, by the pigeonhole principle, there exists some $I' \subseteq I$ with $|I'| \leq 2^s$ which captures at least $\frac{1}{|I|^{2^s}}$ of the total mass of $\nu_{u,I|S}$:

$$\sum_{\substack{x_u,x_I,x_S \\ I^*(x_u,x_I|x_S)=I'}} \frac{\mathbb{P}(X_S = x_S)}{2^{|I|+1}} \nu_{u,I|S}(x_u, x_I|x_S) \geq \frac{1}{|I|^{2^s}} \nu_{u,I|S}.$$

Applying Lemma 9 to each of the terms $\nu_{u,I|S}(x_u, x_I|x_S)$ that we sum over on the left-hand side, we get

$$\sum_{\substack{x_u,x_I,x_S \\ I^*(x_u,x_I|x_S)=I'}} \frac{\mathbb{P}(X_S = x_S)}{2^{|I|+1}} \nu_{u,I'|S}(x_u, x_{I'}|x_S) \geq \frac{1}{(4|I|)^{2^s}} \left( \frac{1}{|I|} \right)^{2^s(2^s+1)} \nu_{u,I|S}.$$

Note that we also have

$$\nu_{u,I'|S} = \sum_{x_u,x_{I'},x_S} \frac{\mathbb{P}(X_S = x_S)}{2^{|I'|+1}} \nu_{u,I'|S}(x_u, x_{I'}|x_S)$$

$$= 2^{|I|-|I'|} \sum_{x_u,x_{I'},x_S} \frac{\mathbb{P}(X_S = x_S)}{2^{|I|+1}} \nu_{u,I'|S}(x_u, x_{I'}|x_S)$$

$$\geq \sum_{\substack{x_u,x_I,x_S \\ I^*(x_u,x_I|x_S)=I'}} \frac{\mathbb{P}(X_S = x_S)}{2^{|I|+1}} \nu_{u,I'|S}(x_u, x_{I'}, x_S).$$

The inequality step above holds because, for each assignment $x_{I'}$, there are $2^{|I|-|I'|}$ assignments $x_I$ that are in accord with it. Hence, each term $\nu_{u,I'|S}(x_u, x_{I'}, x_S)$ can appear at most $2^{|I|-|I'|}$ times in the sum on the last line. Therefore,

$$\nu_{u,I'|S} \geq \sum_{\substack{x_u, x_I, x_S \\ q(x_u, x_I | x_S) = I'}} \frac{\mathbb{P}(X_S = x_S)}{2^{|I|+1}} \nu_{u,I'|S}(x_u, x_{I'}, x_S) \geq \frac{1}{(4|I|)^{2^s}} \left(\frac{1}{|I|}\right)^{2^s(2^s+1)} \nu_{u,I|S}.$$

$\square$

# B  Proof of Theorem 6

Most of the results in this section are restatements of results in [11], with small modifications. Hence, most of the proofs in this section reuse the language of the proofs in [11] verbatim.

Let $A$ be the event that for all $u$, $I$, and $S$ with $|I| \leq 2^s$ and $|S| \leq L$ simultaneously, $|\nu_{u,I|S} - \hat{\nu}_{u,I|S}| < \tau'/2$. Then Lemma 18 gives a result on the number of samples required for event $A$ to hold.

**Lemma 18** (Corollary of Lemma 5.3 in [11]). *If the number of samples is larger than*

$$\frac{60 \cdot 2^{2L}}{(\tau')^2(e^{-2\gamma})^{2L}} \left(\log(1/\omega) + \log(L + 2^s + 1) + (L + 2^s + 1)\log(2n) + \log 2\right),$$

*then* $\mathbb{P}(A) \geq 1 - \omega$.

Now, Lemmas 19-21 provide the ingredients necessary to prove correctness, assuming that event $A$ holds.

**Lemma 19** (Analogue of Lemma 5.4 in [11]). *Assume that the event $A$ holds. Then every time variables are added to $S$ in Step 2 of the algorithm, the mutual information $I(X_u; X_S)$ increases by at least $(\tau')^2/8$.*

*Proof.* Following the proof of Lemma 5.4 in [11], we have that when event $A$ holds,

$$\sqrt{\frac{1}{2} \cdot I(X_u; X_I | X_S)} \geq \frac{1}{2}\nu_{u,I|S} \geq \frac{1}{2}(\hat{\nu}_{u,I|S} - \tau'/2).$$

The algorithm only adds variables to $S$ if $\hat{\nu}_{u,I|S} > \tau'$, so

$$I(X_u; X_I | X_S) \geq \frac{1}{2}(\hat{\nu}_{u,I|S} - \tau'/2)^2 \geq \frac{1}{2}(\tau' - \tau'/2)^2 = (\tau')^2/8.$$

$\square$

**Lemma 20** (Analogue of Lemma 5.5 in [11]). *Assume that the event $A$ holds. Then at the end of Step 2 $S$ contains all of the neighbors of $u$.*

*Proof.* Following the proof of Lemma 5.5 in [11], we have that Step 2 ended either because $|S| > L$ or because there was no set of variables $I \subseteq [n] \setminus (\{u\} \cup S)$ with $\hat{\nu}_{u,I|S} > \tau'$.

If $|S| > L$, we have by Lemma 19 that $I(X_u; X_S) > L \cdot (\tau')^2/8 = 1$. However, because $X_u$ is a binary variable, we also have $1 \geq H(X_u) \geq I(X_u; X_S)$, so we obtain a contradiction.

Suppose then that $|S| \leq L$, but that there was no set of variables $I \subset [n] \setminus (\{u\} \cup S)$ with $|I| \leq 2^s$ and $\hat{\nu}_{u,I|S} > \tau'$. If $S$ does not contain all of the neighbors of $u$, then we know by Theorem 5 that there exists a set of variables $I \subseteq [n] \setminus (\{u\} \cup S)$ with $|I| \leq 2^s$ with $\nu_{u,I|S} \geq 2\tau'$. Because event $A$ holds, we know that $\hat{\nu}_{u,I|S} \geq \nu_{u,I|S} - \tau'/2 > \tau'$. This contradicts our supposition that there was no such set of variables.

Therefore, $S$ must contain all of the neighbors of $u$. $\square$

**Lemma 21** (Analogue of Lemma 5.6 in [11]). *Assume that the event $A$ holds. If at the start of Step 3 $S$ contains all of the neighbors of $u$, then at the end of Step 3 the remaining set of variables are exactly the neighbors of $u$.*

*Proof.* Following the proof of Lemma 5.6 in [11], we have that if event $A$ holds, then

$$\hat{\nu}_{u,i|S\setminus\{i\}} < \nu_{u,i|S\setminus\{i\}} + \tau'/2 \leq \sqrt{\frac{1}{2}I(X_u; X_i | X_S)} + \tau'/2 = \tau'/2$$

for all variables $i$ that are not neighbors of $u$. Then all such variables are pruned. Furthermore, by Theorem 5,

$$\hat{\nu}_{u,i|S\setminus\{i\}} \geq \nu_{u,i|S\setminus\{i\}} - \tau'/2 \geq 2\tau' - \tau'/2 > \tau'$$

for all variables $i$ that are neighbors of $u$, and thus no neighbor is pruned. $\square$

*Proof of Theorem 6 (Analogue of Theorem 5.7 in [11]).* Event $A$ occurs with probability $1 - \omega$ for our choice of $M$. By Lemmas 20 and 21, the algorithm returns the correct set of neighbors of $u$ for every observed variable $u$.

To analyze the running time, observe that when running the algorithm at an observed variable $u$, the bottleneck is Step 2, in which there are at most $L$ steps and in which the algorithm must loop over all subsets of vertices in $[n] \setminus \{u\} \setminus S$ of size $2^s$, of which there are $\sum_{l=1}^{2^s} \binom{n}{l} = O(n^{2^s})$ many. Running the algorithm at all observed variables thus takes $O(MLn^{2^s+1})$ time. $\qquad\square$

# C   Structure Learning Algorithm of Section 5

The steps of the structure learning algorithm are:

1. Fix parameters $s$, $\tau'(\zeta \cdot \eta)$, $\tau'(\zeta)$, $L$. Fix observed variable $u$. Set $S := \emptyset$.

2. While $|S| \leq L$ and there exists a set of observed variables $I \subseteq [n] \setminus \{u\} \setminus S$ of size at most $2^s$ such that $\hat{\nu}_{u,I|S} > \tau'(\zeta \cdot \eta)$, set $S := S \cup I$.

3. For each $i \in S$, if $\hat{\nu}_{u,I|S\setminus\{i\}} < \tau'(\zeta)$ for all sets of observed variables $I \subseteq [n]\setminus\{u\}\setminus(S\setminus\{i\})$ of size at most $2^s$, mark $i$ for removal from $S$.

4. Remove from $S$ all variables marked for removal.

5. Return set $S$ as an estimate of the neighborhood of $u$.

In the algorithm above, we use

$$L = 8/(\tau'(\zeta \cdot \eta))^2, \quad \eta = \frac{1}{\sqrt{3} \cdot 2^{D/2+2^s} \cdot D^{2^{s-1}(2^s+2)}},$$

$$\tau'(x) = \frac{1}{(4d)^{2^s}} \left(\frac{1}{d}\right)^{2^s(2^s+1)} \tau(x), \text{ and } \tau(x) = \frac{1}{2} \frac{4x^2(e^{-2\gamma})^{d+D-1}}{d^{4d}2^{d+1}\binom{D}{d-1}\gamma e^{2\gamma}}.$$

The main difference in the analysis of this algorithm compared to that of the algorithm in Section 3 is that, at the end of Step 2, $S$ is no longer guaranteed to contain all the neighbors of $u$. Then, a smaller threshold is used in Step 2 compared to Step 3 in order to ensure that $S$ contains enough neighbors of $u$ such that the mutual information proxy with any non-neighbor is small.

# D Proof of Theorem 11

See Appendix C for a detailed description of the structure learning algorithm in Section 5.

The correctness of the algorithm is based on the results in Theorem 22 and Lemma 23, which are analogues of Theorem 5 and Lemma 21. We state these, and then we prove Theorem 11 based on them. Then, Section D.1 proves Theorem 22 and Section D.2 proves Lemma 23.

**Theorem 22** (Analogue of Theorem 5). *Fix an observed variable $u$ and a subset of observed variables $S$, such that the two are disjoint. Suppose there exists a neighbor $i$ of $u$ not contained in $S$ such that the MRF of the observed variables contains a Fourier coefficient associated with both $i$ and $u$ that has absolute value at least $\zeta$. Then there exists some subset $I$ of the MRF neighborhood of $u$ with $|I| \le 2^s$ such that*

$$\nu_{u,I|S} \ge \frac{1}{(4d)^{2^s}} \left(\frac{1}{d}\right)^{2^s(2^s+1)} \frac{4\zeta^2(e^{-2\gamma})^{d+D-1}}{d^{4d}2^{d+1}\binom{D}{d-1}\gamma e^{2\gamma}} = 2\tau'(\zeta).$$

Let $A_{\zeta,\eta}$ be the event that for all $u$, $I$, and $S$ with $|I| \le 2^s$ and $|S| \le L$ simultaneously, $|\nu_{u,I|S} - \hat{\nu}_{u,I|S}| < \tau'(\zeta \cdot \eta)/2$.

**Lemma 23** (Analogue of Lemma 21). *Assume that the event $A_{\zeta,\eta}$ holds. If at the start of Step 3 $S$ contains all of the neighbors of $u$ which are connected to $u$ through a Fourier coefficient of absolute value at least $\zeta \cdot \eta$, then at the end of Step 4 the remaining set of variables is a subset of the neighbors of $u$, such that all neighbors which are connected to $u$ through a Fourier coefficient of absolute value at least $\zeta$ are included in the subset.*

*Proof of Theorem 11.* Event $A_{\zeta,\eta}$ occurs with probability $1 - \omega$ for our choice of $M$. Then, based on the result of Theorem 22, we have that Lemmas 18, 19, and 20 hold exactly as before, with $\tau'(\zeta \cdot \eta)$ instead of $\tau'$, and with the guarantee that at the end of Step 2 $S$ contains all of the neighbors of $u$ wihch are connected to $u$ through a Fourier coefficient of absolute value at least $\zeta \cdot \eta$. Finally, Lemma 23 guarantees that the pruning step results in the desired set of neighbors for every observed variable $u$.

The analysis of the running time is identical to that in Theorem 6. □

## D.1 Proof of Theorem 22

We will argue that Theorem 4.6 in [11] holds in the following modified form, which only requires the existence of one Fourier coefficient that has absolute value at least $\alpha$:

**Theorem 24** (Modification of Theorem 4.6 in [11]). *Fix a vertex $u$ and a subset of the vertices $S$ which does not contain the entire neighborhood of $u$, and assume that there exists an $\alpha$-nonvanishing hyperedge containing $u$ and which is not contained in $S \cup \{u\}$. Then taking $I$ uniformly at random from the subsets of the neighbors of $u$ not contained in $S$ of size $s = \min(r - 1, |\Gamma(u) \setminus S|)$,*

$$\mathbb{E}_I\left[\sqrt{\frac{1}{2}I(X_u; X_I|X_S)}\right] \ge \mathbb{E}_I[\nu_{u,I|S}] \ge C'(\gamma, K, \alpha)$$

*where explicitly*

$$C'(\gamma, K, \alpha) := \frac{4\alpha^2\delta^{r+d-1}}{r^{4r}K^{r+1}\binom{D}{r-1}\gamma e^{2\gamma}}.$$

Then, this allows us to prove Theorem 22 with a proof nearly identical to that of Theorem 5.

*Proof of Theorem 22.* Using Theorem 24, we get that there exists some subset $I$ of neighbors of $u$ with $|I| \le d - 1$ such that

$$\nu_{u,I|S} \ge \frac{4\zeta^2(e^{-2\gamma})^{d+D-1}}{d^{4d}2^{d+1}\binom{D}{d-1}\gamma e^{2\gamma}} = 2\tau(\zeta).$$

Then, using Theorem 4, we have that there exists some subset $I' \subseteq I$ with $|I'| \le 2^s$ such that

$$\nu_{u,I'|S} \ge \frac{1}{(4d)^{2^s}} \left(\frac{1}{d}\right)^{2^s(2^s+1)} \frac{4\zeta^2 (e^{-2\gamma})^{d+D-1}}{d^{4d} 2^{d+1} \binom{D}{d-1} \gamma e^{2\gamma}} = 2\tau'(\zeta).$$

$\square$

What remains is to show that Theorem 24 is true. Theorem 24 differs from Theorem 4.6 in [11] only in that it requires at least one hyperedge containing $u$ and not contained in $S \cup \{u\}$ to be $\alpha$-nonvanishing, instead of requiring all maximal hyperedges to be $\alpha$-nonvanishing. The proof of Theorem 4.6 in [11] uses the fact that all maximal hyperedges are $\alpha$-nonvanishing in exactly two places: Lemma 3.3 and Lemma 4.5. In both of these lemmas, it can be easily shown that the same result holds even if only one, not necessarily maximal, hyperedge is $\alpha$-nonvanishing. In fact, the original proofs of these lemmas do not make use of the assumption that all maximal hyperedges are $\alpha$-nonvanishing: they only use that there exists a maximal hyperedge that is $\alpha$-nonvanishing.

We now reprove Lemma 3.3 and Lemma 4.5 in [11] under the new assumption. These proofs contain only small modifications compared to the original proofs. Hence, most of the content of these proofs is restated, verbatim, from [11].

Lemma 25 is a trivial modification of Lemma 2.7 in [11], to allow the tensor which is lower bounded in absolute value by a constant $\kappa$ to be non-maximal. Then, Lemma 26 is the equivalent of Lemma 3.3 in [11] and Lemma 27 is the equivalent of Lemma 4.5 in [11], under the assumption that there exists at least one hyperedge containing $u$ that is $\alpha$-nonvanishing.

**Lemma 25** (Modification of Lemma 2.7 in [11]). *Let $T^{1\cdots s}$ be a centered tensor of dimensions $d_1 \times \ldots \times d_s$ and suppose there exists at least one entry of $T^{1\cdots s}$ which is lower bounded in absolute value by a constant $\kappa$. For any $1 \le l \le r$, and $i_1 < \ldots < i_l$ such that $\{i_1, \ldots, i_l\} \ne [s]$, let $T^{i_1 \cdots i_l}$ be an arbitrary centered tensor of dimensions $d_{i_1} \times \ldots \times d_{i_l}$. Let*

$$T(a_1, \ldots, a_r) = \sum_{l=1}^{r} \sum_{i_1 < \ldots < i_l} T^{i_1 \cdots i_l}(a_{i_1}, \ldots, a_{i_l}).$$

*Then there exists an entry of $T$ of absolute value lower bounded by $\kappa/s^s$.*

*Proof.* Suppose all entries of $T$ are less than $\kappa/s^s$ in absolute value. Then by Lemma 2.6 in [11], all the entries of $T^{1\cdots s}$ are less than $\kappa$ in absolute value. This is a contradiction, so there exists an entry of $T$ of absolute value lower bounded by $\kappa/s^s$. $\square$

**Lemma 26** (The statement is the same as that of Lemma 3.3 in [11]).

$$\mathbb{E}_{Y,Z}\left[\sum_R \sum_{B \ne R} \left(\mathcal{E}_{u,R}^Y - \mathcal{E}_{u,B}^Y - \mathcal{E}_{u,R}^Z + \mathcal{E}_{u,B}^Z\right)\left(\exp(\mathcal{E}_{u,R}^Y + \mathcal{E}_{u,B}^Z) - \exp(\mathcal{E}_{u,B}^Y + \mathcal{E}_{u,R}^Z)\right)\right]$$
$$\ge \frac{4\alpha^2 \delta^{r-1}}{r^{2r} e^{2\gamma}}.$$

*Proof under relaxed $\alpha$ assumption.* Following the original proof of Lemma 3.3, set $a = \mathcal{E}_{u,R}^Y + \mathcal{E}_{u,B}^Z$ and $b = \mathcal{E}_{u,B}^Y + \mathcal{E}_{u,R}^Z$, and let $D' = K^3 \exp(2\gamma) \ge D$. Then we have

$$\mathbb{E}_{Y,Z}\left[\sum_R \sum_{B \ne R} (a-b)(e^a - e^b)\right] = \mathbb{E}\left[\sum_R \sum_{B \ne R} (a-b) \int_b^a e^x dx\right]$$

$$\ge \mathbb{E}\left[\sum_R \sum_{B \ne R}(a-b)^2 e^{-2\gamma}\right] \ge \frac{1}{e^{2\gamma}} \sum_R \sum_{B \ne R} \text{Var}[a-b].$$

By Claim 3.4 in [11], we have

$$\sum_R \sum_{R \ne B} \text{Var}[a-b] = 4k_u \sum_R \text{Var}[\mathcal{E}_{u,R}^Y].$$

Select a hyperedge $J = \{u, j_1, ..., j_s\}$ containing $u$ with $|J| \leq r$, such that $\theta^{uJ}$ is $\alpha$-nonvanishing. Then we get, for some fixed choice $Y_{\sim J}$,

$$\sum_R \text{Var}[\mathcal{E}_{u,R}^Y] \geq \sum_R \text{Var}[\mathcal{E}_{u,R}^Y | Y_{\sim J}] = \sum_R \text{Var}[T(R, Y_{j_1}, ..., Y_{j_s}) | Y_{\sim J}]$$

where the tensor $T$ is defined by treating $Y_{\sim J}$ as fixed as follows:

$$T(R, Y_{j_1}, ..., Y_{j_s}) = \sum_{l=2}^{r} \sum_{i_2 < ... < i_l} \theta^{u i_2 ... i_l}(R, Y_{i_2}, ..., Y_{i_l}).$$

From Lemma 25, it follows that $T$ is $\alpha/r^r$-nonvanishing. Then there is a choice of $R$ and $G$ such that $|T(R, G)| \geq \alpha/r^r$. Because $T$ is centered there must be a $G'$ so that $T(R, G')$ has the opposite sign, so $|T(R, G) - T(R, G')| \geq \alpha/r^r$. Then we have

$$\text{Var}[T(R, Y_{j_1}, ..., Y_{j_s}) | Y_{\sim J}] \geq \frac{\alpha^2 \delta^{r-1}}{2 r^{2r}}$$

which follows from the fact that $\mathbb{P}(Y_{J \setminus u} = G)$ and $\mathbb{P}(Y_{J \setminus u} = G')$ are both lower bounded by $\delta^{r-1}$, and by then applying Claim 3.5 in [11]. Overall, then,

$$\mathbb{E}_{Y,Z}\left[\sum_R \sum_{B \neq R} (a - b)\left(e^a - e^b\right)\right] \geq \frac{4\alpha^2 \delta^{r-1}}{r^{2r} e^{2\gamma}}.$$

$\square$

**Lemma 27** (The statement is the same as that of Lemma 4.5 in [11]). *Let $E$ be the event that conditioned on $X_S = x_S$ where $S$ does not contain all the neighbors of $u$, node $u$ is contained in at least one $\alpha/r^r$-nonvanishing hyperedge. Then $\mathbb{P}(E) \geq \delta^d$.*

*Proof under relaxed $\alpha$ assumption.* Following the original proof of Lemma 4.5, when we fix $X_S = x_S$ we obtain a new MRF where the underlying hypergraph is

$$\mathcal{H}' = ([n] \setminus S, H'), \quad H' = \{h \setminus S | h \in H\}.$$

Let $\phi(h)$ be the image of a hyperedge $h$ in $\mathcal{H}$ in the new hypergraph $\mathcal{H}'$.

Let $h^*$ be a hyperedge in $\mathcal{H}$ that contains $u$ and is $\alpha$-nonvanishing. Let $f_1, ..., f_p$ be the preimages of $\phi(h^*)$ so that without loss of generality $f_1$ is $\alpha$-nonvanishing. Let $J = \cup_{i=1}^p f_i \setminus \{u\}$. Finally let $J_1 = J \cap S = \{i_1, i_2, ..., i_s\}$ and let $J_2 = J \setminus S = \{i_1', i_2', ..., i_{s'}'\}$. We define

$$T(R, a_1, ..., a_s, a_1', ..., a_{s'}') = \sum_{i=1}^{p} \theta^{f_i}$$

which is the clique potential we get on hyperedge $\phi(h^*)$ when we fix each index in $J_1 \subseteq S$ to their corresponding value.

Because $\theta^{f_1}$ is $\alpha$-nonvanishing, it follows from Lemma 25 that $T$ is $\alpha/r^r$-nonvanishing. Thus there is some setting $a_1^*, ..., a_s^*$ such that the tensor

$$T'(R, a_1', ..., a_{s'}') = T(R, a_1^*, ..., a_s^*, a_1', ..., a_{s'}')$$

has at least one entry with absolute value at least $\alpha/r^r$. What remains is to lower bound the probability of this setting. Since $J_1$ is a subset of the neighbors of $u$ we have $|J_1| \leq d$. Thus the probability that $(X_{i_1}, ..., X_{i_s}) = (a_1^*, ..., a_s^*)$ is bounded below by $\delta^s \geq \delta^d$, which completes the proof. $\square$

### D.2 Proof of Lemma 23

The proof of Lemma 21 does not generalize to the setting of Lemma 23 because at the end of Step 2 $S$ is no longer guaranteed to contain the entire neighborhood of $u$.

Instead, the proof of Lemma 23 is based on the following observation: any $\nu_{u,I|S}$, where $I$ is a set of non-neighbors of $u$, is upper bounded within some factor of $\nu_{u,n^*(u) \setminus S|S}$, where $n^*(u)$ is the set of

neighbors of $u$. Intuitively, this follows because any information between $u$ and $I$ must pass through the neighbors of $u$. Then, by guaranteeing that $\nu_{u,n^*(u)\setminus S|S}$ is small, we can also guarantee that $\nu_{u,I|S}$ is small. This allows us to guarantee that all non-neighbors of $u$ are pruned.

Lemma 28 makes formal a version of the upper bound on the mutual information proxy mentioned above. Then, we prove Lemma 23.

**Lemma 28.** *Let $X \in \mathcal{X}, Y \in \mathcal{Y}, Z \in \mathcal{Z}, S \in \mathcal{S}$ be discrete random variables. Suppose $X$ is conditionally independent of $Z$, given $(Y, S)$. Then*

$$\nu_{X,Z|S} \leq \frac{|\mathcal{Y}|}{|\mathcal{Z}|}\nu_{X,Y|S}.$$

*Proof.*

$$\nu_{X,Z|S} = \mathbb{E}_S \sum_{x\in\mathcal{X}} \frac{1}{|\mathcal{X}|} \sum_{z\in\mathcal{Z}} \frac{1}{|\mathcal{Z}|} |\mathbb{P}(X=x, Z=z|S) - \mathbb{P}(X=x|S)\mathbb{P}(Z=z|S)|$$

$$= \mathbb{E}_S \sum_{x\in\mathcal{X}} \frac{1}{|\mathcal{X}|} \sum_{z\in\mathcal{Z}} \frac{1}{|\mathcal{Z}|}$$

$$\cdot \left| \sum_{y\in\mathcal{Y}} (\mathbb{P}(X=x, Y=y, Z=z|S) - \mathbb{P}(X=x|S)\mathbb{P}(Y=y, Z=z|S)) \right|$$

$$\leq \mathbb{E}_S \sum_{x\in\mathcal{X}} \frac{1}{|\mathcal{X}|} \sum_{z\in\mathcal{Z}} \frac{1}{|\mathcal{Z}|}$$

$$\cdot \sum_{y\in\mathcal{Y}} |\mathbb{P}(X=x, Y=y, Z=z|S) - \mathbb{P}(X=x|S)\mathbb{P}(Y=y, Z=z|S)|$$

$$\overset{(*)}{=} \mathbb{E}_S \sum_{x\in\mathcal{X}} \frac{1}{|\mathcal{X}|} \sum_{z\in\mathcal{Z}} \frac{1}{|\mathcal{Z}|}$$

$$\cdot \sum_{y\in\mathcal{Y}} \mathbb{P}(Z=z|Y=y, S) |\mathbb{P}(X=x, Y=y|S) - \mathbb{P}(X=x|S)\mathbb{P}(Y=y|S)|$$

$$= \frac{|\mathcal{Y}|}{|\mathcal{Z}|} \mathbb{E}_S \sum_{x\in\mathcal{X}} \frac{1}{|\mathcal{X}|} \sum_{y\in\mathcal{Y}} \frac{1}{|\mathcal{Y}|} |\mathbb{P}(X=x, Y=y|S) - \mathbb{P}(X=x|S)\mathbb{P}(Y=y|S)|$$

$$= \frac{|\mathcal{Y}|}{|\mathcal{Z}|} \nu_{X,Y|S}$$

where in (*) we used that $\mathbb{P}(Z=z|X=x, Y=y, S) = \mathbb{P}(Z=z|Y=y, S)$, because $Z$ is conditionally independent of $X$, given $(Y, S)$. $\square$

*Proof of Lemma 23.* Consider any $i \in S$ such that $i$ is not a neighbor of $u$, and let $I$ with $|I| \leq 2^s$ be any subset of $[n] \setminus \{u\} \setminus (S \setminus \{i\})$. Let $I^*$ be the set of neighbors of $u$ not included in $S$. Note that $u$ is conditionally independent of $I$, given $(I^*, S \setminus \{i\})$. Then, by Lemma 28,

$$\nu_{u,I|S\setminus\{i\}} \leq \frac{2^{|I^*|}}{2^{|I|}} \nu_{u,I^*|S\setminus\{i\}} \leq 2^{D-1} \nu_{u,I^*|S\setminus\{i\}}.$$

By Lemma 10, there exists a subset $I^\dagger \subseteq I^*$ with $|I^\dagger| \leq 2^s$ such that

$$\nu_{u,I^\dagger|S\setminus\{i\}} \geq \frac{1}{(4|I^*|)^{2^s}} \left(\frac{1}{|I^*|}\right)^{2^s(2^s+1)} \nu_{u,I^*|S\setminus\{i\}} \geq \frac{1}{(4D)^{2^s}} \left(\frac{1}{D}\right)^{2^s(2^s+1)} \nu_{u,I^*|S\setminus\{i\}}.$$

Then, putting together the two results above,

$$\nu_{u,I|S\setminus\{i\}} \leq 2^{D-1}(4D)^{2^s} D^{2^s(2^s+1)} \nu_{u,I^\dagger|S\setminus\{i\}}.$$

Note that

$$\nu_{u,I^\dagger|S\setminus\{i\}} \leq \hat{\nu}_{u,I^\dagger|S\setminus\{i\}} + \tau'(\zeta \cdot \eta)/2 \overset{(*)}{\leq} \tau'(\zeta \cdot \eta) + \tau'(\zeta \cdot \eta)/2 = 3\tau'(\zeta \cdot \eta)/2$$

where in (*) we used that, if $\hat{\nu}_{u,I^{\dagger}|S\setminus\{i\}}$ were larger than $\tau'(\zeta \cdot \eta)$, the algorithm would have added $I^{\dagger}$ to $S$. Then

$$
\begin{aligned}
\nu_{u,I|S\setminus\{i\}} &\leq 3 \cdot 2^{D-2}(4D)^{2^s} D^{2^s(2^s+1)}\tau'(\zeta \cdot \eta) \\
&= \eta^2 \cdot 3 \cdot 2^{D-2}(4D)^{2^s} D^{2^s(2^s+1)}\tau'(\zeta) \\
&= \tau'(\zeta)/4
\end{aligned}
$$

where we used that $\tau'(\zeta \cdot \eta) = \eta^2 \tau'(\zeta)$ and then we replaced $\eta$ by its definition. Putting it all together,

$$
\hat{\nu}_{u,I|S\setminus\{i\}} \leq \nu_{u,I|S\setminus\{i\}} + \tau'(\zeta \cdot \eta)/2 \leq \tau'(\zeta)/4 + \eta^2 \tau'(\zeta)/2 < \tau'(\zeta)
$$

where we used that $\eta \leq 1$. Therefore, all variables $i \in S$ which are not neighbors of $u$ are pruned.

Consider now variables $i$ which are connected to $u$ through a Fourier coefficient of absolute value at least $\zeta$. We know that all variables connected through a Fourier coefficient at least $\zeta \cdot \eta$ are in $S$, so all variables $i$ must also be in $S$, because $\eta \leq 1$. Then, by Theorem 22, there exists a subset $I$ of $[n] \setminus \{u\} \setminus (S \setminus \{i\})$ with $|I| \leq 2^s$, such that

$$
\hat{\nu}_{u,I|S\setminus\{i\}} \geq \nu_{u,I|S\setminus\{i\}} - \tau'(\zeta \cdot \eta)/2 \geq \nu_{u,I|S\setminus\{i\}} - \tau'(\zeta)/2 \overset{(\dagger)}{\geq} 2\tau'(\zeta) - \tau'(\zeta)/2 > \tau'(\zeta)
$$

where in (†) we used the guarantee of Theorem 22, knowing that there exists a variable in $[n] \setminus \{u\} \setminus (S \setminus \{i\})$ connected to $u$ through a Fourier coefficient of absolute value at least $\zeta$: specifically, variable $i$. Therefore, no variables $i \in S$ which are connected to $u$ through a Fourier coefficient of absolute value at least $\zeta$ are pruned. $\qquad\square$

# E   Proof of Theorem 12

Let $\psi$ be the maximum over observed variables of the number of non-zero potentials that include that variable:

$$\psi := \max_{u \in [n]} \sum_{\substack{S \subseteq [n] \\ u \in S}} \mathbb{1}\{\hat{f}(S) \neq 0\}.$$

Theorem 29, stated below, is a stronger version of Theorem 12, in which the upper bound on $\zeta$ depends on $\psi$ instead of $D^d$. This section proves Theorem 29. Note that $\psi \leq \sum_{k=0}^{d-1} \binom{D}{k} \leq D^{d-1} + 1 < D^d$, so Theorem 29 immediately implies Theorem 12.

**Theorem 29.** *Fix $\delta > 0$ and $\epsilon > 0$. Suppose that we are given neighborhoods $n(u)$ for every observed variable $u$ satisfying the guarantees of Theorem 11. Suppose that we are given $M$ samples from the RBM, and that we have*

$$M = \Omega\left(\gamma^2 \ln(8 \cdot n \cdot 2^D / \delta)/\epsilon^2\right), \quad \zeta \leq \frac{\sqrt{\epsilon}}{\psi\sqrt{1 + e^{2\gamma}}}.$$

*Let $z_u$ and $\hat{w}_u$ be the features and the estimate of the weights when the regression algorithm is run at observed variable $u$. Then, with probability at least $1 - \delta$, for all variables $u$,*

$$\mathbb{E}\left[\left(\mathbb{P}(X_u = 1 | X_{\backslash u}) - \sigma\left(\hat{w}_u \cdot z_u\right)\right)^2\right] \leq \epsilon.$$

Define the empirical risk and the risk, respectively:

$$\hat{\mathcal{L}}(w) = \frac{1}{M} \sum_{i=1}^{M} l(y^{(i)}(w \cdot z^{(i)})), \quad \mathcal{L}(w) = \mathbb{E}[l(y(w \cdot z))].$$

Following is an outline of the proof of Theorem 29. Lemma 34 bounds the KL divergence between the true predictor and the predictor that uses $\bar{w}$, where $\bar{w} \in \mathbb{R}^{2^{|n(s)|}}$ is the vector of true weights for every subset of $n(u)$, multiplied by two. Unfortunately, the estimate $\hat{w}$ that optimizes the empirical risk will typically not recover the true weights, because $n(u)$ is not the true set of neighbors of $u$. Lemma 33 decomposes the KL divergence between the true predictor and the predictor that uses $\hat{w}$ in terms of $\mathcal{L}(\hat{w}) - \mathcal{L}(\bar{w})$ and the KL divergence that we bounded in Lemma 34. The term $\mathcal{L}(\hat{w}) - \mathcal{L}(\bar{w})$ can be shown to be small through concentration arguments, which are partially given in Lemma 30. Thus, we obtain a bound on the KL divergence between the true predictor and the predictor that uses $\hat{w}$. Finally, using Lemma 31, we bound the mean-squared error of interest in terms of this KL divergence.

We now give the lemmas mentioned above and complete formally the proof of Theorem 12.

**Lemma 30.** *With probability at least $1 - \rho$ over the samples, we have for all $w \in \mathbb{R}^{2^{|n(u)|}}$ such that $||w||_1 \leq 2\gamma$,*

$$\mathcal{L}(w) \leq \hat{\mathcal{L}}(w) + 4\gamma\sqrt{\frac{2\ln(2 \cdot 2^D)}{M}} + 2\gamma\sqrt{\frac{2\ln(2/\rho)}{M}}.$$

*Proof.* We have $||z||_\infty \leq 1$, $y \in \{-1, 1\}$, the loss function is 1-Lipschitz, and our hypothesis set is $w \in \mathbb{R}^{2^{|n(S)|}}$ such that $||w||_1 \leq 2\gamma$. Then the result follows from Lemma 7 in [28]. $\square$

**Lemma 31** (Pinsker's inequality)**.** *Let $D_{KL}(a, b) = a\ln(a/b) + (1 - a)\ln((1 - a)/(1 - b))$ denote the KL divergence between two Bernoulli distributions $(a, 1 - a)$, $(b, 1 - b)$ with $a, b \in [0, 1]$. Then*

$$(a - b)^2 \leq \frac{1}{2}D_{KL}(a||b).$$

**Lemma 32** (Inverse of Pinsker's inequality; see Lemma 4.1 in [10])**.** *Let $D_{KL}(a, b) = a\ln(a/b) + (1 - a)\ln((1 - a)/(1 - b))$ denote the KL divergence between two Bernoulli distributions $(a, 1 - a)$, $(b, 1 - b)$ with $a, b \in [0, 1]$. Then*

$$D_{KL}(a, b) \leq \frac{1}{\min(b, 1 - b)}(a - b)^2.$$

**Lemma 33.** *For any $w \in \mathbb{R}^{2^{|n(u)|}}$ with $||w||_1 \leq 2\gamma$, we have that*

$$\mathcal{L}(\hat{w}) - \mathcal{L}(w) = \mathbb{E}_z \left[ D_{KL} \left( \frac{\mathbb{E}[y|z] + 1}{2}, \sigma(\hat{w} \cdot z) \right) - D_{KL} \left( \frac{\mathbb{E}[y|z] + 1}{2}, \sigma(w \cdot z) \right) \right].$$

*Proof.*

$$\begin{aligned}
\mathcal{L}(\hat{w}) - \mathcal{L}(w) &= \mathbb{E}_{z,y} \left[ -\frac{y+1}{2} \ln \sigma(\hat{w} \cdot z) - \frac{1-y}{2} \ln(1 - \sigma(\hat{w} \cdot z)) \right] \\
&\quad - \mathbb{E}_{z,y} \left[ -\frac{y+1}{2} \ln \sigma(w \cdot z) - \frac{1-y}{2} \ln(1 - \sigma(w \cdot z)) \right] \\
&= \mathbb{E}_z \left[ -\frac{\mathbb{E}[y|z] + 1}{2} \ln \sigma(\hat{w} \cdot z) - \frac{1 - \mathbb{E}[y|z]}{2} \ln(1 - \sigma(\hat{w} \cdot z)) \right] \\
&\quad - \mathbb{E}_z \left[ -\frac{\mathbb{E}[y|z] + 1}{2} \ln \sigma(w \cdot z) - \frac{1 - \mathbb{E}[y|z]}{2} \ln(1 - \sigma(w \cdot z)) \right] \\
&= \mathbb{E}_z \left[ \frac{\mathbb{E}[y|z] + 1}{2} \ln \frac{\sigma(w \cdot z)}{\sigma(\hat{w} \cdot z)} + \frac{1 - \mathbb{E}[y|z]}{2} \ln \frac{1 - \sigma(w \cdot z)}{1 - \sigma(\hat{w} \cdot z)} \right] \\
&= \mathbb{E}_z \left[ D_{KL} \left( \frac{\mathbb{E}[y|z] + 1}{2}, \sigma(\hat{w} \cdot z) \right) - D_{KL} \left( \frac{\mathbb{E}[y|z] + 1}{2}, \sigma(w \cdot z) \right) \right].
\end{aligned}$$

$\square$

**Lemma 34.** *Let $\zeta \leq \frac{\sqrt{\epsilon}}{\psi \sqrt{1 + e^{2\gamma}}}$ and $\bar{w} \in \mathbb{R}^{2^{|n(u)|}}$ with $\bar{w}_S = 2\hat{f}(S)$ for all $S \subseteq n(u)$. Then, for all assignments $z \in \{-1, 1\}^{2^{|n(u)|}}$,*

$$D_{KL} \left( \frac{\mathbb{E}[y|z] + 1}{2}, \sigma(\bar{w} \cdot z) \right) \leq \epsilon.$$

*Proof.* By Lemma 32, we have that

$$D_{KL} \left( \frac{\mathbb{E}[y|z] + 1}{2}, \sigma(\bar{w} \cdot z) \right) \leq \frac{1}{\min(\sigma(\bar{w} \cdot z), 1 - \sigma(\bar{w} \cdot z))} \left( \frac{\mathbb{E}[y|z] + 1}{2} - \sigma(\bar{w} \cdot z) \right)^2.$$

Note that $\mathbb{E}[y|z^*] = 2\sigma(w^* \cdot z^*) - 1$ for $w^*$ and $z^*$ corresponding to the true neighborhood of $u$, and that $||w^*||_1 \leq 2\gamma$. Note that $\bar{w}_S = w_S^*$ for all $S \subseteq n(u)$. Also note that $\min(\sigma(\bar{w} \cdot z), 1 - \sigma(\bar{w} \cdot z)) \geq \sigma(-2\gamma) = \frac{1}{1 + e^{2\gamma}}$. Then:

$$\begin{aligned}
&D_{KL} \left( \frac{\mathbb{E}[y|z] + 1}{2}, \sigma(\bar{w} \cdot z) \right) \\
&\leq (1 + e^{2\gamma}) \left( \frac{\mathbb{E}[y|z] + 1}{2} - \sigma(\bar{w} \cdot z) \right)^2 \\
&\overset{(a)}{=} (1 + e^{2\gamma}) \left( \mathbb{E}_{z^*|z} \left[ \sigma(w^* \cdot z^*) - \sigma(\bar{w} \cdot z) \right] \right)^2 \\
&\overset{(b)}{\leq} (1 + e^{2\gamma}) \mathbb{E}_{z^*|z} \left( \sigma(w^* \cdot z^*) - \sigma(\bar{w} \cdot z) \right)^2 \\
&= (1 + e^{2\gamma}) \mathbb{E}_{z^*|z} \left( \sigma \left( \sum_{S \subseteq n^*(u)} \hat{f}(S \cup \{u\}) \chi_S(x) \right) - \sigma \left( \sum_{S \subseteq n(u)} \hat{f}(S \cup \{u\}) \chi_S(x) \right) \right)^2 \\
&\overset{(c)}{\leq} (1 + e^{2\gamma}) \mathbb{E}_{z^*|z} \left( \sum_{S \subseteq n^*(u)} \hat{f}(S \cup \{u\}) \chi_S(x) - \sum_{S \subseteq n(u)} \hat{f}(S \cup \{u\}) \chi_S(x) \right)^2
\end{aligned}$$

$$= (1 + e^{2\gamma})\mathbb{E}_{z^*|z} \left( \sum_{\substack{S \subseteq n^*(u) \\ S \not\subseteq n(u)}} \hat{f}(S \cup \{u\})\chi_S(x) \right)^2$$

$$\overset{(d)}{\leq} (1 + e^{2\gamma})\psi^2\zeta^2$$

where in (a) we used the law of iterated expectations, in (b) we used Jensen's inequality, in (c) we used that $\sigma$ is 1-Lipschitz, and in (d) we used that the Fourier coefficients that we sum over are all upper bounded in absolute value by $\zeta$ (otherwise the corresponding sets $S$ would need to be included in $n(u)$, by the assumption that $n(u)$ contains all the neighbors connected to $u$ through a Fourier coefficient of absolute value at least $\zeta$). Therefore, setting $\zeta \leq \frac{\sqrt{\epsilon}}{\psi\sqrt{1+e^{2\gamma}}}$ achieves error $\epsilon$. $\qquad\square$

*Proof of Theorem 29.* Let $M \geq C \cdot \gamma^2 \ln(8 \cdot n \cdot 2^D/\delta)/\epsilon^2$, for some global constant $C$. Then, by Lemma 30, with probability at least $1 - \delta/(2n)$, for all $w \in \mathbb{R}^{2^{|n(u)|}}$ such that $||w||_1 \leq 2\gamma$,

$$\mathcal{L}(\hat{w}) \leq \hat{\mathcal{L}}(\hat{w}) + \epsilon/2.$$

Note that $l(y(w \cdot z)) = \ln(1 + e^{y(w \cdot z)})$ is bounded because $|y(w \cdot z)| \leq 2\gamma$, and $|\ln(1+e^{-2\gamma}) - \ln(1 + e^{2\gamma})| \leq 4\gamma$ because the function is 1-Lipschitz. Then, by Hoeffding's inequality, $\mathbb{P}(\hat{\mathcal{L}}(w) - \mathcal{L}(w) \geq t) \leq e^{-2Mt^2/(4\gamma)^2}$. Then, for $M \geq C' \cdot \gamma^2 \ln(2n/\delta)/\epsilon^2$ for some global constant $C'$, with probability at least $1 - \delta/(2n)$,

$$\hat{\mathcal{L}}(w) \leq \mathcal{L}(w) + \epsilon/2.$$

Then the following holds with probability at least $1 - \delta/n$ for any $w \in \mathbb{R}^{2^{|n(u)|}}$ with $||w||_1 \leq 2\gamma$:

$$\mathcal{L}(\hat{w}) \leq \hat{\mathcal{L}}(\hat{w}) + \epsilon/2 \leq \hat{\mathcal{L}}(w) + \epsilon/2 \leq \mathcal{L}(w) + \epsilon.$$

Then we have

$$\mathbb{E}\left[ \left( \mathbb{P}(X_u = 1|X_{[n]\setminus\{u\}}) - \sigma\left(\hat{w} \cdot z\right) \right)^2 \right]$$

$$\overset{(a)}{\leq} \frac{1}{2}\mathbb{E}\left[ D_{KL}\left( \mathbb{P}(X_u = 1|X_{[n]\setminus\{u\}}), \sigma\left(\hat{w} \cdot z\right) \right) \right]$$

$$\overset{(b)}{=} \frac{1}{2}(\mathcal{L}(\hat{w}) - \mathcal{L}(\bar{w})) + \frac{1}{2}\mathbb{E}\left[ D_{KL}\left( \mathbb{P}(X_u = 1|X_{[n]\setminus\{u\}}), \sigma\left(\bar{w} \cdot z\right) \right) \right]$$

$$\overset{(c)}{\leq} \frac{1}{2}(\mathcal{L}(\hat{w}) - \mathcal{L}(\bar{w})) + \frac{1}{2}\epsilon$$

$$\overset{(d)}{\leq} \epsilon$$

where in (a) we used Lemma 31, in (b) we used Lemma 33, in (c) we used Lemma 34, and in (d) we used that $\mathcal{L}(\hat{w}) - \mathcal{L}(\bar{w}) \leq \epsilon$.

By a union bound, this holds for all variables $u$ with probability at least $1 - \delta$. $\qquad\square$

# F   A weaker width suffices

The sample complexity of the algorithm in Section 3 depends on $\gamma$, the width of the MRF of the observed variables. A priori, it is unclear how large $\gamma$ can be. Ideally, we would have an upper bound on $\gamma$ in terms of parameters that are natural to the RBM, such as the width of the RBM $\beta^*$.

In Section F.3, we give an example of an RBM for which $\beta^* = d \ln d$ and $\gamma$ is linear in $\beta^*$ and exponential in $d$. This example shows that a bound $\gamma \leq \beta^*$ between the widths of the MRF and of the RBM does not generally hold.

However, we show that a bound $\gamma^* \leq \beta^*$ holds for a modified width $\gamma^*$ of the MRF. Then, we show that $\gamma$ can be replaced with $\gamma^*$ everywhere in the analysis of our algorithm (and of that of [11]), without any other change in its guarantees.

$\gamma^*$ is always less than or equal to $\gamma$, and, as we discussed, sometimes strictly less than $\gamma$. Hence, the former dependency on $\gamma$ was suboptimal. By replacing $\gamma$ with $\gamma^*$, we improve the sample complexity, and we make the dependency interpretable in terms of the width of the RBM.

## F.1   Main result

Let $\gamma^*$ be the modified width of an MRF, defined as

$$\gamma^* := \max_{u \in [n]} \max_{I \subseteq [n] \setminus \{u\}} \max_{x \in \{-1,1\}^n} \left| \sum_{S \subseteq I} \hat{f}(S \cup \{u\}) \chi_{S \cup \{u\}}(x) \right|.$$

Whereas $\gamma$ is a sum of absolute values of Fourier coefficients, $\gamma^*$ requires the signs of the Fourier coefficients that it sums over to be consistent with some assignment $x \in \{-1, 1\}^n$. Note that it is always the case that $\gamma^* \leq \gamma$.

Lemma 35 shows that $\gamma^* \leq \beta^*$. Then, in Section F.2 we argue that $\gamma$ and $\gamma^*$ are interchangeable for the guarantees of the algorithm in [11], and implicitly for the guarantees of the algorithm in Section 3. Finally, in Section F.3 we give an example of an RBM for which $\gamma$ is linear in $\beta^*$ and exponential in $d$.

**Lemma 35.** *Consider an RBM with width $\beta^*$, and let $\gamma^*$ be the modified width of the MRF of the observed variables. Then $\gamma^* \leq \beta^*$.*

*Proof.* We have

$$\mathbb{P}(X_u = x_u | X_{[n] \setminus \{u\}} = x_{[n] \setminus \{u\}}) = \frac{\exp\left( \sum_{\substack{S \subseteq [n] \\ u \in S}} \hat{f}(S) \chi_S(x) \right)}{\exp\left( -\sum_{\substack{S \subseteq [n] \\ u \in S}} \hat{f}(S) \chi_S(x) \right) + \exp\left( \sum_{\substack{S \subseteq [n] \\ u \in S}} \hat{f}(S) \chi_S(x) \right)}$$

$$= \sigma\left( 2 \sum_{\substack{S \subseteq [n] \\ u \in S}} \hat{f}(S) \chi_S(x) \right).$$

On the other hand, we have

$$\sigma(-2\beta^*) \leq \mathbb{P}(X_u = x_u | X_{[n] \setminus \{u\}} = x_{[n] \setminus \{u\}}) \leq \sigma(2\beta^*).$$

Therefore, by the monotonicity of the sigmoid function, we have for all $x \in \{-1, 1\}^n$,

$$-\beta^* \leq \sum_{\substack{S \subseteq [n] \\ u \in S}} \hat{f}(S) \chi_S(x) \leq \beta^*,$$

or equivalently,

$$-\beta^* \leq \sum_{S \subseteq [n] \setminus \{u\}} \hat{f}(S \cup \{u\}) \chi_{S \cup \{u\}}(x) \leq \beta^*.$$

Denote $\phi(x_1, ..., x_n) = \sum_{S \subseteq [n] \setminus \{u\}} \hat{f}(S \cup \{u\}) \chi_{S \cup \{u\}}(x)$. Then the following marginalization result holds for any $i \neq u$:

$$\sum_{S \subseteq [n] \setminus \{u,i\}} \hat{f}(S \cup \{u\}) \chi_{S \cup \{u\}}(x)$$
$$= \frac{\phi(x_1, ..., x_{i-1}, -1, x_{i+1}, ..., x_n) + \phi(x_1, ..., x_{i-1}, 1, x_{i+1}, ..., x_n)}{2}.$$

Because the lower bound $-\beta$ and upper bound $\beta$ apply to each $\phi(x_1, ..., x_n)$, we get that the same bounds apply to the marginalized value:

$$-\beta^* \leq \sum_{S \subseteq [n] \setminus \{u,i\}} \hat{f}(S \cup \{i\}) \chi_{S \cup \{i\}}(x) \leq \beta^*.$$

This marginalization result extends trivially to marginalizing multiple variables. Then, by marginalizing all variables $x_i$ for $i \notin I \cup \{u\}$ for some $I \subseteq [n] \setminus \{u\}$, we get the bounds

$$-\beta^* \leq \sum_{S \subseteq I} \hat{f}(S \cup \{u\}) \chi_{S \cup \{u\}}(x) \leq \beta^*.$$

Taking the maximum over $u \in [n]$, $I \in [n] \setminus \{u\}$, and $x \in \{-1, 1\}^n$, we get that $\gamma^* \leq \beta^*$. $\qquad \square$

### F.2 The same guarantees hold with the weaker width

For the algorithm in Section 3, the dependence on $\gamma$ comes only from the use of Theorem 5, for which the dependence on $\gamma$ comes only from the use of Theorem 4.6 in [11]. Hence, it is sufficient to show that Theorem 4.6 in [11] admits the same guarantees when $\gamma$ is replaced with $\gamma^*$.

The modifications that need to be made to the proof of Theorem 4.6 in [11] are trivial: it is sufficient to replace every occurence of the symbol $\gamma$ with the symbol $\gamma^*$. This is because the proof does not use any property of $\gamma$ that is not also a property of $\gamma^*$.

In the rest of this section, we briefly review the occurences of $\gamma$ in the proof of Theorem 4.6 in [11] and argue that they can be replaced with $\gamma^*$. Toward this goal, the rest of this section will use the notation of [11]. We direct the reader to that paper for more information.

We first define $\gamma^*$ in the setting of [11]. We have

$$\gamma^* := \max_{u \in [n]} \max_{I \in [n] \setminus \{u\}} \max_{X_1 \in [k_1], ..., X_n \in [k_n]} \left| \sum_{l=1}^{r} \sum_{i_2 < ... < i_l} \mathbb{1}_{\{i_2 ... i_l\} \subseteq I} \theta^{u i_2 ... i_l}(X_u, X_{i_2}, ..., X_{i_l}) \right|$$

and

$$\delta^* := \frac{1}{K} \exp(-2\gamma^*).$$

With these definitions, for any variable $X_u$ and assignment $R$, we have for its neighborhood $X_U$ that

$$\mathbb{P}(X_u = R | X_U) \geq \frac{\exp(-\gamma^*)}{K \exp(\gamma^*)} = \frac{1}{K} \exp(-2\gamma^*) = \delta^*.$$

Similarly to [11], we also have that that if we pick any variable $X_i$ and consider the new MRF given by conditioning on a fixed assignment of $X_i$, then the value of $\gamma^*$ for the new MRF is non-increasing.

$\gamma$ and $\delta$ appear in the proof of Theorem 4.6 in [11] as part of Lemma 3.1, Lemma 3.3, Lemma 4.1, and Lemma 4.5. We now aruge, for each of these lemmas, that $\gamma$ and $\delta$ can be replaced with $\gamma^*$ and $\delta^*$, respectively.

**Lemma 3.1 in [11]**. $\gamma$ is used as part of the upper bound $|\Phi(R, I, X_i)| \leq \gamma \binom{D}{r-1}$, which is used to conclude that the total amount wagered is at most $\gamma K \binom{D}{r-1}$. The upper bound follows from the derivation

$$|\Phi(R, I, X_i)| = \left| \sum_{l=1}^{s} C_{u,l,s} \sum_{i_1 < i_2 < ... < i_l} \mathbb{1}_{\{i_1 ... i_l\} \subseteq I} \theta^{u i_1 ... i_l}(R, X_{i_1}, ..., X_{i_l}) \right|$$

$$\leq \binom{D}{r-1} \left| \sum_{l=1}^{s} \sum_{i_1 < i_2 < ... < i_l} \mathbb{1}_{\{i_1...i_l\} \subseteq I} \theta^{u i_1 ... i_l}(R, X_{i_1}, ..., X_{i_l}) \right|$$

$$\leq \gamma \binom{D}{r-1}.$$

By the definition of $\gamma^*$, the second inequality holds exactly the same with $\gamma^*$, so we also get that $|\Phi(R, I, X_i)| \leq \gamma^* \binom{D}{r-1}$. Then, the total amount wagered is at most $\gamma^* K \binom{D}{r-1}$.

**Lemma 3.3 in [11].** This lemma gives a lower bound of $\frac{4\alpha^2 \delta^{r-1}}{r^{2r} e^{2\gamma}}$ on an expectation of interest. We want to replace $\delta$ with $\delta^*$ in the numerator and $\gamma$ with $\gamma^*$ in the denominator.

For the numerator, $\delta$ comes from the lower bounds $\mathbb{P}(Y_{J \setminus u} = G) \geq \delta^{r-1}$ and $\mathbb{P}(Y_{J \setminus u} = G') \geq \delta^{r-1}$. Note that $Y$ is identical in distribution to $X$, the vector of random variables of the MRF. Let $S \subseteq [n]$, $i \in S$, and let $n^*(i)$ denote the set of neighbors of variable $X_i$. Then the lower bounds mentioned above come from the following marginalization argument:

$$\mathbb{P}(X_S = x_S) = \mathbb{P}(X_i = x_i | X_{S \setminus i} = x_{S \setminus i}) \mathbb{P}(X_{S \setminus i} = x_{S \setminus i})$$

$$= \left( \sum_{x_{n^*(i) \setminus S}} \mathbb{P}(X_i = x_i | X_{n^*(i) \cap S} = x_{n^*(i) \cap S}, X_{n^*(i) \setminus S} = x_{n^*(i) \setminus S}) \right.$$

$$\left. \cdot \mathbb{P}(X_{n^*(i) \setminus S} = x_{n^*(i) \setminus S} | X_{S \setminus i} = x_{S \setminus i}) \right) \mathbb{P}(X_{S \setminus i} = x_{S \setminus i})$$

$$\geq \left( \sum_{x_{n^*(i) \setminus S}} \delta \cdot \mathbb{P}(X_{n^*(i) \setminus S} = x_{n^*(i) \setminus S} | X_{S \setminus i} = x_{S \setminus i}) \right) \mathbb{P}(X_{S \setminus i} = x_{S \setminus i})$$

$$= \delta \cdot \mathbb{P}(X_{S \setminus i} = x_{S \setminus i}).$$

By applying the bound recursively, we obtain $\mathbb{P}(X_S = x_S) \geq \delta^{|S|}$. Then, because $|J \setminus u| \leq r - 1$, we get the desired lower bound of $\delta^{r-1}$. Note, however, that the inequality step in the derivation above also holds for $\delta^*$, as it only uses that $\mathbb{P}(X_u = R | X_U) \geq \delta^*$. Therefore, we can use $(\delta^*)^{r-1}$ in the numerator.

For the denominator, $e^{2\gamma}$ comes from the lower bounds $\mathcal{E}_{u,R}^Y + \mathcal{E}_{u,B}^Z \geq -2\gamma$ and $\mathcal{E}_{u,B}^Y + \mathcal{E}_{u,R}^Z \geq -2\gamma$. Recall that

$$\mathcal{E}_{u,R}^X = \sum_{l=1}^{r} \sum_{i_2 < ... < i_l} \theta^{u i_2 ... i_l}(R, X_{i_2}, ..., X_{i_l}).$$

Then, by the definition of $\gamma^*$, these lower bounds also hold trivially with $\gamma^*$, so we can use $e^{2\gamma^*}$ in the denominator.

**Lemma 4.1 in [11].** In this lemma, $\gamma$ appears in an upper bound of $\gamma K \binom{D}{r-1}$ on the total amount wagered. We showed in Lemma 3.1 that the total amount wagered is at most $\gamma^* K \binom{D}{r-1}$, so we can use $\gamma^*$ instead of $\gamma$.

**Lemma 4.5 in [11].** In this lemma, $\delta$ appears in the lower bound $\mathbb{P}(X_{i_1} = a_1^*, ..., X_{i_s} = a_s^*) \geq \delta^s$, which also holds with $\delta^*$ instead of $\delta$ by the argument that $\mathbb{P}(X_S = x_s) \geq (\delta^*)^{|S|}$ that we developed in our description of Lemma 3.3.

Therefore, it is possible to reaplce $\gamma$ with $\gamma^*$ and $\delta$ with $\delta^*$ everywhere in the proof of Theorem 4.6 in [11], and implicitly also in all the proofs of the the algorithm in Section 3.

### F.3 Example of RBM with large width

This section gives an example of an RBM with width linear in $\beta^*$ and exponential in $d$. The RBM consists of a single latent variable connected to $d$ observed variables. There are no external fields, and all the interactions have the same value. Note that, in this case, each interaction is equal to $\frac{\beta^*}{d}$, where $\beta^*$ is the width of the RBM.

For this RBM, the MRF induced by the observed variables has a probability mass function

$$\mathbb{P}(X = x) \propto \exp\left(\rho\left(\frac{\beta^*}{d}(x_1 + ... + x_d)\right)\right).$$

The analysis of $\gamma$ for this MRF is based on the fact that, for large arguments, the function $\rho$ is well approximated by the absolute value function, for which the Fourier coefficients can be explicitly calculated.

Lemma 36 gives a lower bound on the "width" corresponding to the Fourier coefficients of the absolute value function applied to $x_1 + ... + x_d$. Then, Lemma 37 gives a lower bound on $\gamma$ for the RBM described above, in the case when $\beta^* \geq d\ln d$. This lower bound is linear in $\beta^*$ and exponential in $d$.

**Lemma 36.** *Let $g : \{-1, 1\}^d \to \mathbb{R}$ with $g(x) = |x_1 + ... + x_d|$. Let $\hat{g}$ be the Fourrier coefficients of $g$. Then, for $d$ multiple of $4$ plus $1$, for all $u \in [d]$,*

$$\sum_{\substack{S \subseteq [d] \\ u \in S}} |\hat{g}(S)| \geq \frac{2^{(d-1)/2}}{2\sqrt{d-1}}.$$

*Proof.* Note that, for $x \in \{-1, 1\}^d$, we have

$$|x_1 + ... + x_d| = \text{Maj}_d(x_1, ..., x_d) \cdot (x_1 + ... + x_d)$$

where $\text{Maj}_d(x_1, ..., x_d)$ is the majority function, equal to 1 if more than half of the arguments are 1 and equal to $-1$ otherwise. Because $d$ is odd, the definition is non-ambiguous. The Fourier coefficients of $\text{Maj}_d(x_1, ..., x_d)$ are known to be (see Chapter 5.3 in [22]):

$$\hat{\text{Maj}}_d(S) = \begin{cases} (-1)^{(|S|-1)/2} \frac{1}{2^{d-1}} \binom{d-1}{(d-1)/2} \frac{\binom{(d-1)/2}{(|S|-1)/2}}{\binom{d-1}{|S|-1}} & \text{if } |S| \text{ odd} \\ 0 & \text{if } |S| \text{ even} \end{cases}$$

Let $h_i(x_1, ..., x_d) = \text{Maj}_d(x_1, ..., x_d) \cdot x_i$. The Fourier coefficients $\hat{h}_i$ are obtained from the Fourier coefficients $\hat{\text{Maj}}_d$ by observing the effect of the multiplication by $x_i$: for a set $S$ such that $i \in S$, we get $\hat{h}_i(S) = \hat{\text{Maj}}_d(S \setminus \{i\})$, and for a set $S$ such that $i \notin S$, we get $\hat{h}_i(S) = \hat{\text{Maj}}_d(S \cup \{i\})$. That is:

$$\hat{h}_i(S) = \begin{cases} (-1)^{(|S|-2)/2} \frac{1}{2^{d-1}} \binom{d-1}{(d-1)/2} \frac{\binom{(d-1)/2}{(|S|-2)/2}}{\binom{d-1}{|S|-2}} & \text{if } |S| \text{ even and } i \in S \\ (-1)^{(|S|)/2} \frac{1}{2^{d-1}} \binom{d-1}{(d-1)/2} \frac{\binom{(d-1)/2}{|S|/2}}{\binom{d-1}{|S|}} & \text{if } |S| \text{ even and } i \notin S \\ 0 & \text{if } |S| \text{ odd} \end{cases}$$

Then $\hat{g}$ is simply obtained as $\hat{h}_1 + ... + \hat{h}_d$. This gives:

$$\hat{g}(S) = \begin{cases} (-1)^{(|S|-2)/2} \frac{1}{2^{d-1}} \binom{d-1}{(d-1)/2} \left(|S| \cdot \frac{\binom{(d-1)/2}{(|S|-2)/2}}{\binom{d-1}{|S|-2}} - (d-|S|) \cdot \frac{\binom{(d-1)/2}{|S|/2}}{\binom{d-1}{|S|}}\right) & \text{if } |S| \text{ even} \\ 0 & \text{if } |S| \text{ odd} \end{cases}$$

We will now develop a lower bound for $\hat{g}(S)$ when $|S|$ is even with $|S| > 0$. Using the fact that $\binom{a}{b} = \binom{a}{b+1}\frac{b+1}{a-b}$, we have that when $|S|$ is even with $|S| > 0$,

$$\frac{\binom{(d-1)/2}{(|S|-2)/2}}{\binom{d-1}{|S|-2}} = \frac{\binom{(d-1)/2}{|S|/2}}{\binom{d-1}{|S|}} \cdot \frac{|S|/2}{(d-|S|+1)/2} \cdot \frac{d-|S|+1}{|S|-1} \cdot \frac{d-|S|}{|S|}$$

$$= \frac{\binom{(d-1)/2}{|S|/2}}{\binom{d-1}{|S|}} \cdot \frac{d-|S|}{|S|-1}.$$

Then, when $|S|$ is even with $|S| > 0$,

$$\hat{g}(S) = (-1)^{(|S|-2)/2} \frac{1}{2^{d-1}} \binom{d-1}{(d-1)/2} \frac{\binom{(d-1)/2}{|S|/2}}{\binom{d-1}{|S|}} \left(|S| \frac{d-|S|}{|S|-1} - (d-|S|)\right)$$

$$= (-1)^{(|S|-2)/2} \frac{1}{2^{d-1}} \binom{d-1}{(d-1)/2} \frac{\binom{(d-1)/2}{|S|/2}}{\binom{d-1}{|S|}} \frac{d-|S|}{|S|-1}.$$

Consider the ratio $\frac{|\hat{g}(S)|}{|\hat{g}(S')|}$ for $|S'| = |S| - 2$:

$$\frac{|\hat{g}(S)|}{|\hat{g}(S')|} = \frac{|S|-1}{d-|S|} \frac{\frac{d-|S|}{|S|-1}}{\frac{d-|S|+2}{|S|-3}} = \frac{|S|-3}{d-|S|+2}.$$

This ratio is greater than 1 for $|S| > (d-1)/2 + 3$ and is less than 1 for $|S| < (d-1)/2 + 3$. Because we are only interested in $|S|$ even, we see that the largest value of $|S|$ for which the ratio is less than 1 is $(d-1)/2 + 2$. Hence, $|\hat{g}(S)|$ is minimized at $|S| = (d-1)/2 + 2$ when considering $|S|$ even with $|S| > 0$. (The calculation above is not valid for the case $|S| = 2$ and $|S'| = 0$; however, it is easy to verify explicitly that in that case we have $\frac{|\hat{g}(S)|}{|\hat{g}(S')|} = \frac{1}{d} \leq 1$, so the argument holds.)

It is easy to verify explicitly that at $|S| = (d-1)/2 + 2$ we have

$$|\hat{g}(S)| = \frac{1}{2^{d-1}} \binom{(d-1)/2}{(d-1)/4}.$$

Then this is a lower bound on all $|\hat{g}(S)|$ where $|S|$ is even with $|S| > 0$. Then,

$$|\hat{g}(S)| \geq \frac{1}{2^{d-1}} \binom{(d-1)/2}{(d-1)/4} \overset{(*)}{\geq} \frac{1}{2^{d-1}} \frac{2^{(d-1)/2}}{\sqrt{d-1}} = \frac{1}{\sqrt{d-1} \cdot 2^{(d-1)/2}}$$

where in (*) we used the central binomial coefficient lower bound $\binom{2n}{n} \geq \frac{4^n}{\sqrt{4n}}$.

Then, for any $u \in [d]$,

$$\sum_{\substack{S \subseteq [d] \\ u \in S}} |\hat{g}(S)| \geq 2^{d-2} \cdot \frac{1}{\sqrt{d-1} \cdot 2^{(d-1)/2}} = \frac{2^{(d-1)/2}}{2\sqrt{d-1}}$$

where we used that the number of subsets $S \subseteq [d]$ with $u \in S$ and with $|S|$ even is $2^{d-2}$. $\quad\square$

**Lemma 37.** *For any $d \geq 5$ multiple of 4 plus 1 and $\beta^* \geq d \ln d$, there exists an RBM of width $\beta^*$ with $d$ observed variables and one latent variable such that, in the MRF of the observed variables,*

$$\gamma \geq \beta^* \cdot \frac{2^{(d-1)/2}}{4d^{3/2}}.$$

*Proof.* Let $f(x) = \rho\left(\frac{\beta^*}{d}(x_1 + ... + x_d)\right)$. Then, for the RBM with one latent variable connected to $d$ observed variables through interactions of value $\frac{\beta}{d}$, we have that

$$\mathbb{P}(X = x) \propto \exp(f(x)).$$

Note that this RBM has width $\beta^*$.

Let $g(x) = \left|\frac{\beta^*}{d}(x_1 + ... + x_d)\right|$. Then, if $\hat{f}$ and $\hat{g}$ are the Fourier coefficients corresponding to $f$ and $g$, respectively, we have

$$\|\hat{f} - \hat{g}\|_2^2 \overset{(a)}{=} \frac{1}{2^d} \sum_{x \in \{-1,1\}^d} (f(x) - g(x))^2$$

$$\overset{(b)}{\leq} \left(\rho\left(\frac{\beta^*}{d}\right) - \frac{\beta^*}{d}\right)^2$$

$$= \left(\log(e^{\beta^*/d}(1 + e^{-2\beta^*/d})) - \frac{\beta^*}{d}\right)^2$$

$$= \left(\log(1 + e^{-2\beta^*/d})\right)^2$$

$$\stackrel{(c)}{\leq} e^{-4\beta^*/d}$$

where in (a) we used Praseval's identity, in (b) we used that $(\rho(y) - |y|)^2$ is largest when $|y|$ is smallest and that $\left|\frac{\beta^*}{d}(x_1 + ... + x_d)\right| \geq \frac{\beta^*}{d}$ because $d$ is odd, and in (c) we used that $\log(1 + x) \leq x$. Then

$$||\hat{f} - \hat{g}||_1 \leq 2^{d/2}||\hat{f} - \hat{g}||_2 \leq 2^{d/2}e^{-2\beta^*/d}.$$

Note that the Fourier coefficients of $g(x) = \left|\frac{\beta^*}{d}(x_1 + ... + x_d)\right| = \frac{\beta^*}{d}|x_1 + ... + x_d|$ are $\frac{\beta^*}{d}$ times the Fourier coefficients of $|x_1 + ... + x_d|$. Then, by applying Lemma 36, we have that

$$\max_{u \in [d]} \sum_{\substack{S \subseteq [d] \\ u \in S}} |\hat{f}(S)| \geq \max_{u \in [d]} \sum_{\substack{S \subseteq [d] \\ u \in S}} |\hat{g}(S)| - 2^{d/2}e^{-2\beta^*/d} \geq \frac{\beta^*}{d} \cdot \frac{2^{(d-1)/2}}{2\sqrt{d}} - 2^{d/2}e^{-2\beta^*/d}.$$

We solve for $\beta^*$ such that the second term is at most half the first term. After some manipulations, we get that

$$2^{d/2}e^{-2\beta^*/d} \leq \frac{1}{2}\frac{\beta^*}{d} \cdot \frac{2^{(d-1)/2}}{2\sqrt{d}} \iff \beta^* \geq \frac{5}{4}d\ln 2 + \frac{3}{4}d\ln d - \frac{1}{2}d\ln \beta^*.$$

For $d \geq 5$, it suffices to have $\beta^* \geq d\ln d$. Hence, we obtain

$$\max_{u \in [d]} \sum_{\substack{S \subseteq [d] \\ u \in S}} |\hat{f}(S)| \geq \frac{1}{2}\frac{\beta^*}{d} \cdot \frac{2^{(d-1)/2}}{2\sqrt{d}} = \beta^* \cdot \frac{2^{(d-1)/2}}{4d^{3/2}}.$$

$\square$