[Reviews · NeurIPS 2020]

Review 1

Summary and Contributions: This paper considers the problem of learning a Restricted Boltzmann Machine when the number of latent variables connected to any Markov Random Field neighborhood is small. Specifically, if this number is bounded by s, then the authors give an algorithm to learn the RBM in time O(n^(2^s + 1)), where n is the number of visible nodes in the RBM. This represents an improvement over the current state of the art whenever s < log(d-1), where d is the maximum degree of the latent variables in the RBM. The key ingredient of the argument is a structural result on RBMs, showing that whenever there is a set of nodes with large mutual information with a node, there must exist a subset of those nodes of size 2^s that has large mutual information with the node. Thus, when constructing the neighborhoods of the RBM, it is sufficient to search over subsets of size at most 2^s. The paper also considers the problem of learning such RBMs with respect to prediction error. That is, conditioned on the rest of the nodes, give the probability that a particular node will be set to 1. The authors show that this can be done in much the same way as the other result, with the exception that they do not have any dependence on the minimum potential of the underlying MRF, which can cause a large blow up in the other setting.

Strengths: 1. This paper clearly identifies settings in which they can improve upon the best known time for learning RBMs. They also do a nice job of providing examples where such setting can occur. 2. The paper also identifies an issue with their result (dependence on the minimum potential) and provides a different learning objective where such a dependence can be avoided.

Weaknesses: As this paper builds upon a previously established approach to learning RBMs via MRFs, one may ask whether or not there is enough here that is novel for NeurIPS. I am inclined to believe that this work is novel enough; however, I think that the authors could help their case here by pointing out the difficulties that arise in proving the paper's main structural result that do not arise in other related structural results such as Hamilton et al. (2017).

Correctness: I have not verified all the details of the proofs, but what I have looked at appears to be correct.

Clarity: The paper is clear and well-written.

Relation to Prior Work: This paper does a good job of describing related work, particularly those results which it builds on. As I pointed out in "weaknesses", this paper could be further improved by pointing out the new challenges that arise in proving their structural result that are not present in previous works.

Reproducibility: Yes

Additional Feedback: One question I have about Theorem 11: is it possible to give a similar result when we condition on a smaller subset of nodes (instead of everything except u)? ------------------------------------------------------- After reading the other reviews and the author response, I still feel that my original score is appropriate. I do feel that the authors should include a brief discussion in their paper about the difficulties that arise in proving their structural result (the author response on this point would be a great starting point). This would help reinforce the novelty of the paper and hopefully inspire future work.


Review 2

Summary and Contributions: The authors describe theoretical rates of learning for RBMs with "few latent variables" (connected to any particular visible node). This is a relatively novel task, and their bounds follow by applying some existing work on structure recovery with the addition of a few key insights.

Strengths: * Appears to be a novel task, potentially interesting to the community

Weaknesses: * Precise connection to alternative approaches is unclear (see additional notes) * Significance and usefulness not clear (see additional notes)

Correctness: I think so

Clarity: Mostly. It gives precise definitions, but I found aspects of it confusing (see additional notes).

Relation to Prior Work: Mostly. A notable exception is whether other works could also be applied directly using the authors' "small Markov blanket" assumptions (see notes).

Reproducibility: Yes

Additional Feedback: I found the phrase "few latent variables" to be confusing, since this is not really what the authors mean. In particular, their bounds use "s", which is bounded by the number of latent variables connected to any variable in the Markov blanket of Xi in the marginal distribution over the visible X. This was not clear, in my view, until the formal definition (150-155). Since this style of RBM is not well studied, the practical significance of learning rates is not clear. The type of model is intuitively appealing (many models use "local" latent variables or encourage sparseness), and perhaps the work would spur application of these styles of RBM, but it's difficult to say that this is improving the theory for a well-established problem of importance. In my opinion, the work could also use more intuition about what properties are being leveraged to provide the improved bounds. In particular, one view is that in this setting, we can estimate the Markov blanket of X (which cannot be too big), and then use these cliques to determine the set of latent variables. From that perspective, almost any method that uses independence tests could be applied and achieve "similar" bounds. These would likely not be equivalent, since the authors' method explicitly takes into account the number of latent variables (used in e.g. Lemma 6 to define the mutual information as a weighted sum of 2^s terms), but it is unclear to me how these approaches would be related, or whether the authors' results in better statistical efficiency. Additionally, Section 5 explictly shows how learning such models can be extremely difficult, due to cancellations of correlation in X from different latent variables. The authors' argument appears to be that such errors in the model do not affect the accuracy of conditional predictions, which they formalize using the same bounding techniques. Line 203: connected to observed variables in I : should this also include u?


Review 3

Summary and Contributions: The paper provides an algorithm that correctly recovers the neighborhood of each observed variable, and has better time complexity, for restricted Boltzmann machines with "few latent variables".

Strengths: The theoretical results seem sound.

Weaknesses: Relevance might be the main weakness. (Please see my additional feedback below.)

Correctness: The main claims seem correct.

Clarity: The paper is clear.

Relation to Prior Work: Relation to prior work is properly discussed.

Reproducibility: Yes

Additional Feedback: While the authors properly address the issue that an algorithm better than O(n^d) might not be possible in general (Lines 59-66), a reader might still wonder why the regime s < log(d-1) is relevant for their O(n^{2^s+1}) result to be better than prior work. It might better to motivate this somewhat tangentially from some applied areas. === AFTER REBUTTAL According to the author response, "a bound of d on the maximum degree in an RBM implies that s <= d^3". The authors then assume the regime s < log(d-1). I was expecting some sort of motivation in either theoretical prior work in this or other ML problems, or motivation tangentially from some applied areas. Still, some illustrative examples are given in Figure 1. Given the above, I keep my initial evaluation of 6.

[Author Response · NeurIPS 2020]

We thank the reviewers for their comments.

**Reviewer #1**: Regarding the challenges that arise in proving the structural result: Compared to structural results on learning MRFs, such as [2], the main difference is that we obtain a time complexity for learning RBMs that is believed to be impossible for learning MRFs, by exploiting the latent-variable structure of RBMs. The main challenge is to relate the properties of the induced MRF to this latent-variable structure. The task is non-trvial due to the non-linear effect that latent variables have on observed variables.

To overcome the challenge, the first step is Lemma 6, which is an "interchange of sums" argument that expresses the mutual information on the observed variables as a sum over the latent variables. This expression is convenient in its explicit relation between observed variables and latent variables, but it is still complicated. Then, the second step is Lemma 7, which limits the cancellations that can happen in a sum of products by the number of terms in the sum. Combined, these two steps allow us to constrain the cancellations of the low-order interactions between the observed variables, in terms of the latent-variable structure of the RBM.

Regarding conditioning on smaller subsets of nodes in Theorem 11: Such bounds are plausible, because the conditional probabilities with smaller subsets are simply weighted averages of the full conditional probabilities. However, there is an error both in the terms and in the weights of these weighted averages, and we do not currently have bounds in this direction.

**Reviewer #2**: Regarding the phrase "few latent variables": We agree that the phrase is not sufficiently accurate. Perhaps a better option would be "locally sparse latent variables". We intend to switch to this phrase, or some similar one, in the final version of the paper. We will also ensure that the assumption is made precise earlier in the exposition.

Regarding the properties leveraged to obtain the improved bounds: Some intuition is as follows. The difficulty of estimating the Markov blanket of an observed variable lies in the fact that the low-order interactions between the observed variables can vanish. If that were not the case, a greedy $\tilde{O}(n^2)$ algorithm would work (this is indeed the case if all the RBM weights are positive [1]). Then, our analysis shows that the order up to which all interactions can vanish is constrained by the number of latent variables. Or, in other words, that distributions in which all interactions up to a large order vanish are "complex" and require "many" latent variables.

Regarding other approaches that would estimate the Markov blanket: Unfortunately, estimating the Markov blanket of an observed variable seems to be exactly where the difficulty lies. Based on all prior literature, estimating the Markov blanket would only be guaranteed to scale as $\tilde{O}(n^d)$. Then, our guarantee improves this to $\tilde{O}(n^{2^s+1})$.

Regarding the result in Section 5: In case it was not sufficiently emphasized in the paper, we note that the sample complexity, and hence also the time complexity, of structure learning for RBMs necessarily depends on the minimum potential, due to information-theoretic arguments [3]. All prior algorithms have this dependence. Then, the contribution of Section 5 is that we show a way to eliminate this dependence; we necessarily lose the guarantee on structure recovery, but we still guarantee accurate conditional prediction.

Regarding the definition of $U$ on line 203: The sentence as written is correct. It turns out that it is possible to "factor out" (into $\bar{f}$) the contribution of latent variables that are connected to $u$ but not to any observed variables in $I$. Note that our definition of $s$ also excludes the latent variables that are connected only to the observed variable in question.

**Reviewer #3**: Regarding relevance: Sparsity, in particular as a bound on the maximum degree of a node, is a common assumption in the analysis of graphical models. Note that, by itself, a bound of $d$ on the maximum degree in an RBM implies that $s \leq d^3$. Then, our assumption can be seen as simply a strengthening of this particular aspect of sparsity, such that few latent variables act on the neighborhood of each observed variable.

More generally, our main motivation behind this work was to carve out good sets of assumptions for which efficient learning is possible with time complexity less than $\tilde{O}(n^d)$. This is the first work that achieves such a time complexity without ferromagneticity, i.e., with arbitrary weights in the RBM.

# References

[1] Guy Bresler, Frederic Koehler, and Ankur Moitra. Learning restricted Boltzmann machines via influence maximization. In *Proceedings of the 51st Annual ACM SIGACT Symposium on Theory of Computing*, pages 828–839. ACM, 2019.

[2] Linus Hamilton, Frederic Koehler, and Ankur Moitra. Information theoretic properties of Markov random fields, and their algorithmic applications. In *Advances in Neural Information Processing Systems*, pages 2463–2472, 2017.

[3] Narayana P Santhanam and Martin J Wainwright. Information-theoretic limits of selecting binary graphical models in high dimensions. *IEEE Transactions on Information Theory*, 58(7):4117–4134, 2012.


[Meta-Review · NeurIPS 2020]

This paper presents an algorithm for provably learning RBMs when each visible node is connected to a small number of hiddens, presenting bounds that improve over previous results in a specific regime. While reviewers agree the results appear sound, the paper has done little to convince the reviewers of the significance of the regime, and reviewer requests for additional intuition were not satisfied effectively in the author response. In total, though, the work appears novel and sound, and consensus is in favor of acceptance. I would strongly encourage the authors to try to address R2's questions. (R2's response after rebuttal: "I didn't find the response very helpful, unfortunately. I will have to look at the paper again, but my intuition was that $s$ should bound the size of the Markov blanket, which should lead to better than $O(n^d)$ scaling, and they don't seem to have addressed this except to say it doesn't.")